# Water-oriented magnetic anisotropy transition

Sheng-Qun Su [1], Shu-Qi Wu [1], Masato Hagihala [2,3], Ping Miao [2,4,5], Zhijian Tan [2,4,5], Shuki Torii [2], Takashi Kamiyama [2], Tongtong Xiao [6], Zhenxing Wang [6], Zhongwen Ouyang [6], Yuji Miyazaki [7], Motohiro Nakano [7], Takumi Nakanishi [1], Jun-Qiu Li [1], Shinji Kanegawa [1] & Osamu Sato [1 ✉]

Water reorientation is essential in a wide range of chemical and biological processes. However, the effects of such reorientation through rotation around the metal–oxygen bond on the chemical and physical properties of the resulting complex are usually ignored. Most studies focus on the donor property of water as a recognized σ donor-type ligand rather than a participant in the π interaction. Although a theoretical approach to study water-rotation effects on the functionality of a complex has recently been conducted, it has not been experimentally demonstrated. In this study, we determine that the magnetic anisotropy of a Co(II) complex can be effectively controlled by the slight rotation of coordinating water ligands, which is achieved by a two-step structural phase transition. When the water molecule is rotated by 21.2 ± 0.2° around the Co–O bond, the directional magnetic sus-ceptibility of the single crystal changes by approximately 30% along the *a*-axis due to the rotation of the magnetic anisotropy axis through the modification of the π interaction between cobalt(II) and the water ligand. The theoretical calculations further support the hypothesis that the reorientation of water molecules is a key factor contributing to the magnetic anisotropy transition of this complex.

[1] Institute for Materials Chemistry and Engineering & Integrated Research Consortium on Chemical Sciences (IRCCS), Kyushu University, Fukuoka 819-0395, Japan. [2] Institute of Materials Structure Science, High Energy Accelerator Research Organization (KEK), Tokai, Ibaraki 319-1106, Japan. [3] Department of Materials Structure Science, Sokendai (The Graduate University for Advanced Studies), Tokai, Ibaraki 319-1106, Japan. [4] Institute of High Energy Physics, Chinese Academy of Sciences, Beijing 100049, China. [5] Spallation Neutron Source Science Center, Dongguan 523803, China. [6] Wuhan National High Magnetic Field Center & School of Physics, Huazhong University of Science and Technology, Wuhan 430074, China. [7] Research Center for Thermal and Entropic Science, Graduate School of Science, Osaka University, Toyonaka, Osaka 560-0043, Japan. ✉email: sato@cm.kyushu-u.ac.jp

Water is an important charge-neutral ligand commonly used in coordination complexes. The orientation of water is dependent on hydrogen bonds, which play a vital role in stabilizing the entire structure[1–3]. Due to their small size, high polarity, and hydrogen bonding capability, water molecules readily reorient in response to changes in the substrate and external environments. This phenomenon must be investigated for further application of metal complexes in molecular devices, biological systems, etc. Such reorientation is occasionally also accompanied by a dynamic rearrangement and restructuring of the hydrogen bonding network in the complex[4–6]. Notably, the reorientation of free water is a typical structural perturbation and is widely present in biological systems and at solid–aqueous interfaces. It is also the driving force of various important processes, including the proton transfer and transport, hydration of proteins, enzyme catalysis, and heterogeneous catalysis[7–10]. Nonetheless, the reorientation of water ligands through axial rotation around the metal–oxygen bonds, which might significantly affect the chemical and physical properties of metal complexes, has not been sufficiently evaluated[11,12]. Most studies focus on the coordination polyhedron and bond order of metal–oxygen bonds related to water ligands to elucidate the structure and properties of coordination complexes. This is because the donor property is typically assumed to be dominated by the σ donation rather than the π interaction in most low-valent transition metal complexes. Moreover, the σ donor effect between the water molecule and metal center is not influenced by the rotation around the metal–oxygen bond[13–18]. Recently, theoretical studies have suggested that the easy axis of magnetization in a Dy(III) complex rotated with the rotation of the apical water molecule[14,15]. However, the water reorientation effects have not been confirmed experimentally because controlled water molecule rotation is challenging to achieve, particularly in the densely packed crystalline phase[15,16]. Furthermore, the available methodologies for the detection of the water orientation are limited. Thus, elucidating the reorientation effect of coordinated water in metal complexes is important for their application in various fields, including molecular devices and biomedicine.

The magnetic anisotropy is sensitive to such tiny structural perturbation because, in single-ion transition metal systems, this property typically occurs in the energy range of 10–100 cm$^{-1}$ (ref. [19]), which is comparable to the energy of the weak π ligation energy between the metal center and the water molecule[15]. Therefore, the switching in magnetic anisotropy induced by the rotation of the water ligand around the metal–oxygen bond could be clearly detected. Note that the magnetic anisotropy is of great importance in magnetic materials, since it greatly affects the hysteresis in ferromagnets, slow relaxation of magnetization in single-molecule magnets[2,20], and the operating temperature in quantum computing. Various strategies have been proposed to manipulate magnetic anisotropy, such as tuning the molecular symmetry, lowering the coordination number, and using ligands with soft donor atoms[21–27]. Herein, we describe that a two-step rotational reorientation of a coordinated water molecule in a cobalt(II) complex results in a modification of the electronic structure, with the subsequent substantial changes in magnetic anisotropy. The correlation between the water orientation and magnetic anisotropy is unambiguously determined by single-crystal X-ray diffraction (XRD), neutron powder diffraction, high-frequency/field electron paramagnetic resonance (HF-EPR), and angular-resolved magnetometry studies. The present study shows that the magnetic anisotropy can be controlled through a mere small rotation of the water ligand.

## Results
### Molecular structure, structural transition, and reversible water reorientation.
Cobalt(II) complex, [Co(ONO$_2$)$_2$(H$_2$O)(mprpz)] (**1**)

where mprpz is 2,6-bis(3-methylpyrazol-1-yl)pyrazine (see NMR spectra in Supplementary Fig. 16), was synthesized as described in "Methods". The single-crystal XRD analysis of **1** at 190 K (high-temperature phase, HTp), crystallizes in the triclinic space group *P*-1, where the asymmetric unit cell comprises one complex molecule. The molecule has a six-coordinate structure, where the oxygen from the water molecule is coordinated with the cobalt ion in the equatorial plane formed by mprpz (Fig. 1a). Two nitrate ions in the axial direction adopt the monodentate η$^1$-coordination mode, with Co(II)–O distances ranging from 2.135(1) Å to 2.098(1) Å, and Co(II)–N distances ranging from 2.186(1) Å to 2.111(1) Å (Supplementary Tables 1 and 2). The bond distances between cobalt and the ligand in the high-spin and low-spin states are usually ca. 2.1 and 1.9 Å, respectively. This means that at 190 K, the Co(II) ion in **1** is in the high-spin state[28]. The nitrate ions are twisted along the dihedral angles N3–Co1–O1–O3 ($\psi$) and N3–Co1–O4–O6 ($\omega$) and have the values of 133.53° and 159.64°, respectively (Fig. 1c). The dihedral angle ($\varphi$) between the molecular plane (this plane defined by the N1, N3, and N5 atoms is not easily affected by variations in coordination environment and is close to the Co–O(water) bond (the angle $\varphi_0$ between them is 3.53°)) and the plane of the coordinated water molecule is 85.30° (Fig. 1b). To better describe the orientation of the water, the other parameters angle $\varphi_1$ are introduced. Here the angle $\varphi_1$ between the Co–O(water) bond and the molecular plane of water is 27.56°. Each water molecule is surrounded by four nitrates. The water molecule connects with the adjacent nitrate oxygen through hydrogen bonds, which are essential in the orientation of water molecules. The whole structure is primarily stabilized by O–H···O hydrogen bonds and C–H–π interactions (Fig. 1d and Supplementary Table 3). Heat capacity measurements unambiguously revealed that **1** undergoes a two-step phase transition (Fig. 2 and Supplementary Fig. 1). The phase transition temperature ($T_{trs}$) is determined as 113.5 and 157.4 K. The sharp peaks and the super-cooling phenomenon indicate that a first-order phase transition occurred. To elucidate the crystal

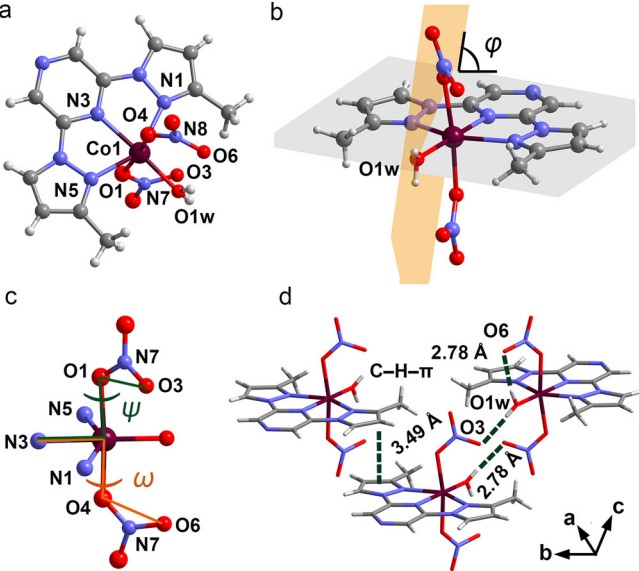

**Fig. 1 Molecular structure and interactions of complex 1. a** Molecular structure of crystal **1** recorded at 190 K. **b** The plane of the coordinated water molecule and the molecular plane form the dihedral angle $\varphi$ with a value of 83.99°. **c** The dihedral angles $\psi$ and $\omega$, which are related to nitrates, are 133.54° and 159.65°, respectively. **d** The entire structure is mainly stabilized by O–H···O hydrogen bonds and C–H–π interactions; the water molecule acts as a hydrogen bond donor. Maroon, Co; gray, C; blue, N; red, O; light gray, H.

structure after the phase transition, single-crystal XRD analysis was performed at 140 and 70 K. The crystal structures at 140 K (intermediate phase, Ip) and 70 K (low-temperature phase, LTp) show that **1** has the same space group as that at 190 K but different unit-cell parameters (Supplementary Fig. 2a and Supplementary Tables 1 and 2). The cobalt centers at 140 K also adopt a similar coordination geometry as those at 190 K with varying degrees of changes in geometric parameters.

The main changes can be observed in the angles $\omega$, $\psi$, $\varphi$, and $\varphi_1$. The angle $\omega$ is 159.64° at 190 K and changes to 153.03° and 141.33° at 140 K, and then to 141.35° at 70 K (Supplementary Fig. 2b and Supplementary Table 4). Meanwhile, the angle $\psi$ is 133.53° at 190 K and changes to 133.49° and 123.80° at 140 K, and then to 126.38° at 70 K. The changes in the angles $\omega$ and $\psi$ result in distinct distortion in the coordination spheres of the Co(II) ions during the phase transition. In addition, the dihedral angle $\varphi$ changes from 85.30° at 190 K to 70.66° and 68.19° at 140 K, and to 61.73° at 70 K (the angle $\varphi_0$ changes slightly from 0.65° at 190 K to 2.22° and 6.07° at 140 K, and to 9.55° at 70 K; Fig. 3 and Supplementary Fig. 2c). The angle $\varphi_1$ changes from 27.56° at 190 K to 30.88° and 21.49° at 140 K, and to 21.22° at 70 K. The changes in $\varphi$ and $\varphi_1$ mean that the water molecule undergoes the reorientation obviously after the phase transition. The orientation of water is dependent on the direction of the hydrogen bonds[29] because the water molecule acts as a donor and forms hydrogen

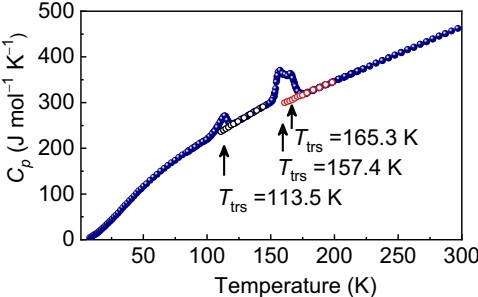

**Fig. 2 Temperature dependence of the heat capacity (**$C_p$**) for 1.** The molar heat capacities under constant pressure ($C_p$) showed two sharp peaks accompanying the latent heat at 113.5 and 157.4 K (there is a shoulder peak at 165.3 K); and exhibited the super-cooling phenomenon. Blue dots, black circles and red circles represent the data obtained during the heating mode after the samples were cooled to 8.0, 110.9, and 160.9 K, respectively.

bonds with nitrates (Supplementary Fig. 3 and Supplementary Table 3). Each water molecule may form hydrogen bonds with four adjacent nitrates. The orientation of water molecules is mainly related to the strongest hydrogen bonds O–H1WA (H2WA)⋯O and O–H1WB(H2WB)⋯O. When the phase transition produces changes in the nitrate's position, the water ligand correspondingly adjusts its orientation to maintain the strongest hydrogen bond interactions. The strongest hydrogen bonds change from O–H1WA⋯O3[$-x+1$, $-y+2$, $-z+1$] and O–H1WB⋯O6 at 190 K to O–H1WA⋯O5[$-x+2$, $-y$, $-z$] and O–H1WB⋯O3[$-x+2$, $-y$, $-z+1$] at 70 K. The electron density ($F_o$–$F_c$) maps[30] of different phases allow to directly observe the reorientation of water (Supplementary Fig. 4). To verify the important roles of hydrogen bonds in the phase transitions, a partially deuterated analog of **1**, [Co(ONO₂)₂(D₂O)(mprpz)] (**1-$d_2$**), was synthesized. The degree of deuteration was estimated to be 28% from the infrared spectrum (Supplementary Fig. 5a, c). As evident in differential scanning calorimetry measurements shown in Supplementary Fig. 5b, the phase transition temperature of the deuterated sample **1-$d_2$** is slightly higher (by approximately 3 K) than that of **1**. The shift in the phase transition temperature following deuteration is consistent with the fact that the hydrogen bonding in the structure substantially affects the orientation of water during the phase transition. To determine the unambiguous orientation of water, the structure of **1** was further analyzed with high-resolution single-crystal X-ray diffraction (HRXRD) (Supplementary Tables 1 and 2). In addition, the structure of the deuterated analog [Co(ONO₂)₂(D₂O)(mprpz-$d_{12}$)] (**1-$d_{14}$**) was analyzed using variable-temperature single-crystal XRD and powder neutron diffraction (PND) (Supplementary Fig. 6 and Supplementary Tables 1 and 5). From four sets of data, the water reorientation in complex **1** is mainly attained by the rotation of water molecules around the Co–O bond after the structural transition. The variation of the angle $\varphi$ in complex **1** from HTp to LTp can be described with the result of HRXRD that is 21.2 ± 0.2° (Supplementary Methods and Supplementary Table 4).

**Anisotropic magnetic susceptibility transition.** Magnetic measurements were performed on both a microcrystalline sample and a single crystal sample of **1** to investigate changes in magnetic properties. As shown in Supplementary Fig. 7a, the susceptibility of the microcrystalline sample decreases with a decrease in the temperature, and an irregularly shaped curve is observed at 70–180 K, which is related to the undulation in the magnetization

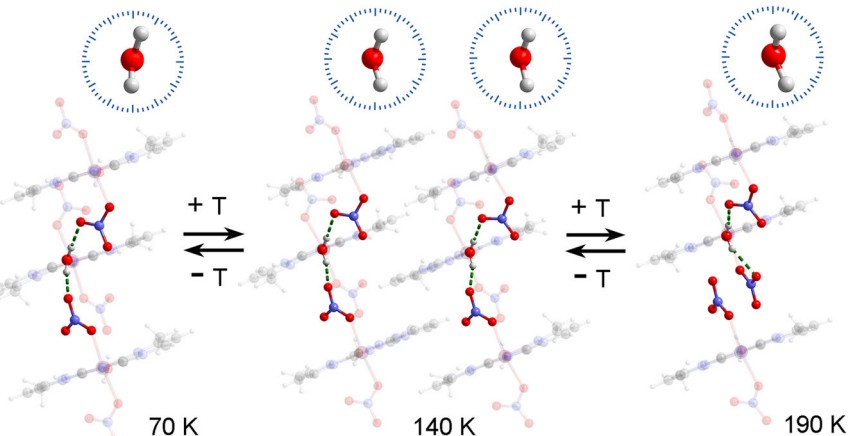

**Fig. 3 Variations in the orientation of the water molecule in complex 1.** The coordinated water molecule undergoes a reversible two-step rotation in response to the temperature, which can be induced by the direction of the hydrogen bonds, O–H⋯O, between the water and nitrates after the structural phase transition. The top images show the orientations of the water molecule at each temperature, where the corresponding dihedral angle, $\varphi$, changes from 61.71° at 70 K to 70.66° and 68.19° at 140 K, and to 85.30° at 190 K. Maroon, Co; gray, C; blue, N; red, O; light gray, H.

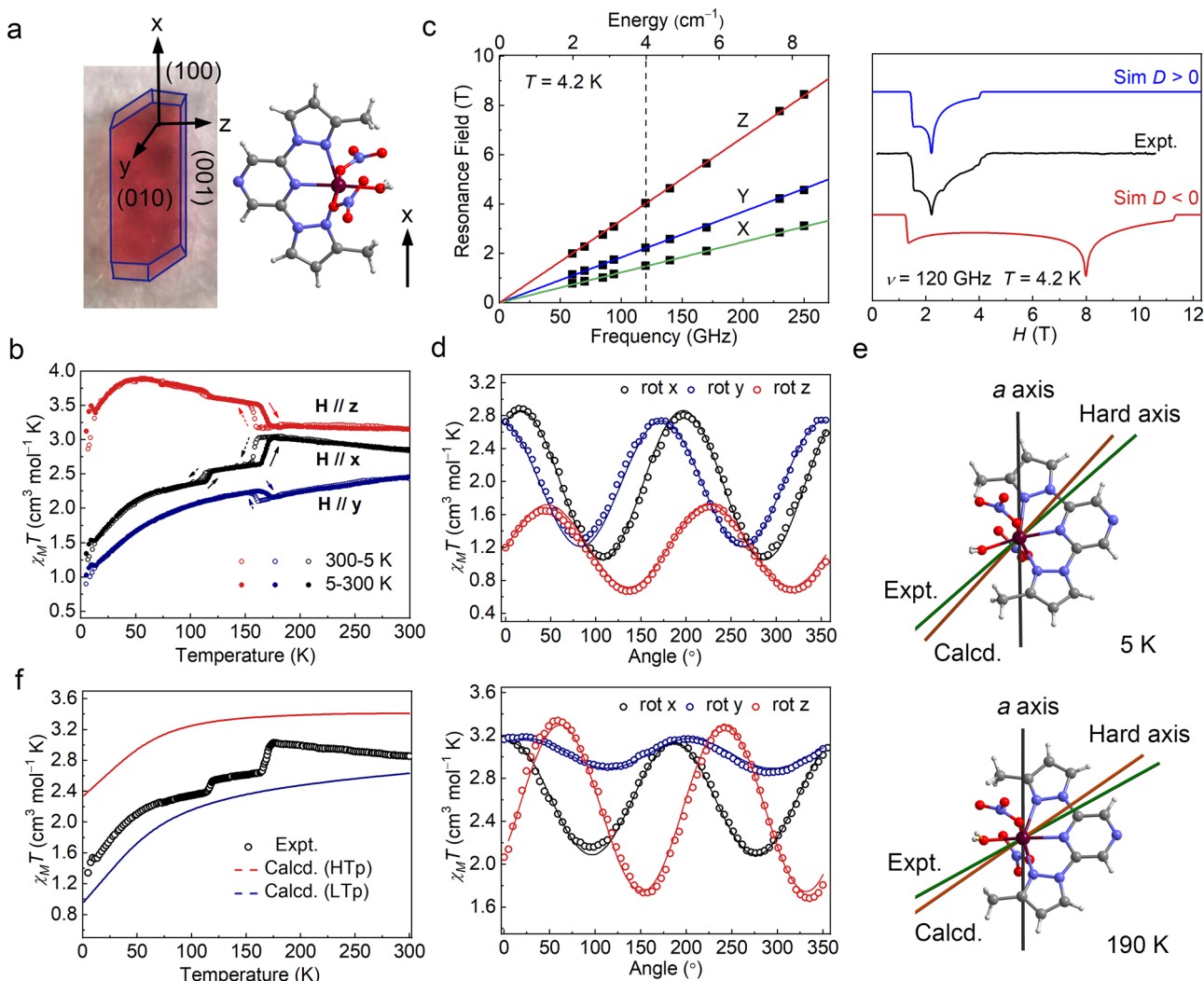

**Fig. 4 Determination of the magneto-crystalline anisotropy and its variations in single crystal 1. a** The shape, faces, and the $x$-, $y$-, and $z$-directions of the single crystal used in magnetic measurements, where the $x$-direction is parallel to the $a$-axis, the $y$-axis is perpendicular to the (010) plane, and the $z$-axis is perpendicular to the $xy$ plane. The relationship between the $x$-axis and the molecular orientation in the crystal is shown. **b** Temperature dependence of the $\chi_M T$ ($\chi_{Mx}T$, $\chi_{My}T$, and $\chi_{Mz}T$) values for single crystal **1**. **c** Left: resonance field versus microwave frequency (quantum energy) for the EPR transitions of **1**, where the green, blue, and red lines correspond to the simulations using the best fit spin Hamiltonian parameters with the magnetic field $H$ parallel to the $X$-, $Y$-, and $Z$-axes of the ZFS tensor, respectively. The vertical dashed line represents the frequency (120 GHz). Right: The HF-EPR spectrum with its simulations at 4.2 K and 120 GHz. **d** Angular dependence of the magnetic susceptibility measured at 5 and 190 K for the rotation along the $x$-, $y$-, and $z$-axes, where the solid lines represent the calculated values. **e** Experimental and ab initio calculated hard axis of the magnetization in crystal **1**. The angle between the experimental hard axis and the $a$-axis changes from 49.8° at 5 K to 61.4° at 190 K. **f** Experimental and ab initio calculated $\chi_{Mx}T$ curves in LTp and HTp. The changes in the calculated values after the phase transition are consistent with the experimental data. Maroon, Co; gray, C; blue, N; red, O; light gray, H.

as a result of a change in the orbital angular momentum owing to the structural transition[31]. The typical Curie–Weiss behavior is observed with the Curie constant of $C = 3.043$ cm$^3$ mol$^{-1}$ K and the Weiss constant of $\theta = -11.735$ K. The effective magnetic moment ($\mu_{eff}$) is calculated to be 4.934 $\mu_B$, which is greater than 3.873 $\mu_B$ for $S = 3/2$, $g = 2$, which indicates an unusually large magnetic anisotropy in the system[32]. The large effective magnetic moment is induced by spin–orbit coupling which partially restores the quenched orbital angular momentum[27]. To carefully examine the anisotropic magnetic properties of **1**, the magnetic properties along the different directions of the single crystal sample were measured. Here, $\chi_{Mx}$, $\chi_{My}$, and $\chi_{Mz}$ represent the molar magnetic susceptibilities when the magnetic field is applied along the $x$-, $y$-, and $z$-directions, respectively. The relationship between $x$-, $y$-, and $z$-directions and the molecular orientation in

the crystal is shown in Fig. 4a and Supplementary Fig. 8a. The molecular orientation is observed to slightly vary after the phase transition, although the Miller indices of the crystal face change. The $\chi_{My}T$ and $\chi_{Mz}T$ values are 2.43 and 3.15 cm$^3$ mol$^{-1}$ K at 300 K, respectively. Upon cooling, these values abruptly increase at around 155 K. Upon heating from 5 K, the $\chi_{My}T$ and $\chi_{Mz}T$ values abruptly decrease at around 165 K, and the temperature-induced hysteresis loop is ca. 10 K. The $\chi_{Mx}T$ value is 2.37 cm$^3$ mol$^{-1}$ K at 300 K. However, in contrast to $\chi_{My}T$ and $\chi_{Mz}T$, upon cooling, the $\chi_{Mx}T$ value abruptly decreases at around 155 K. The $\chi_{Mx}T$ versus T plots also show the hysteresis loop of about 10 K, where the difference in the $\chi_{Mx}T$ values of the two states within the hysteresis loop is about 0.42 cm$^3$ mol$^{-1}$ K. Furthermore, another hysteresis loop with a width of about 7 K is observed at about 110 K, where the change in the $\chi_{Mx}T$ value around this

temperature is about $0.18\,\mathrm{cm^3\,mol^{-1}\,K}$. The overall change in the $\chi_{\mathrm{M}x}T$ value at around 160 and 110 K is $0.60\,\mathrm{cm^3\,mol^{-1}\,K}$ (Fig. 4b). This behavior is completely different from that of the spin transition. It should be noted that the $\chi_{\mathrm{M}y}T$ and $\chi_{\mathrm{M}z}T$ plots also show anomalies at approximately 110 K, which is consistent with the occurrence of the phase transition at approximately 110 K. Consequently, at low temperature, the value of $\chi_{\mathrm{M}z}T$ is much higher than those of $\chi_{\mathrm{M}x}T$ and $\chi_{\mathrm{M}y}T$, whereas the value of $\chi_{\mathrm{M}x}T$ approaches that of $\chi_{\mathrm{M}z}T$ upon heating. The large difference between the susceptibilities along the $x$-, $y$-, and $z$-directions, which persists even up to the room temperature, confirms the unusually large magnetic anisotropy of complex **1**. Furthermore, the field-induced slow relaxation of magnetization is observed, which is directly associated with magnetic anisotropy (Supplementary Fig. 9a–c). For the deuterated sample **1-$d_{14}$**, the temperature dependence of $\chi_{\mathrm{M}x}T$ also exhibits a two-step transition with considerable changes observed at approximately 132 and 173 K (Supplementary Fig. 8b).

**HF-EPR spectroscopy**. To accurately determine the anisotropic parameters of the Co(II) center, a tunable-frequency high-frequency electron paramagnetic resonance (HF-EPR) spectroscopy experiment was performed on a polycrystalline powder sample of **1** at 4.2 K at frequencies between 60 and 250 GHz (Fig. 4c). The full set of spin Hamiltonian parameters, including $D$, $E$, and the values of the $g$-matrix, was obtained by least-squares fitting to the complete two-dimensional array of the resonances. The best fit was obtained with the parameters $(S = 3/2)$: $D = 32.01\,\mathrm{cm^{-1}}$, $|E| = 4.30(2)\,\mathrm{cm^{-1}}$, $g_{(X)} = g_{(Y)} = 2.45(2)$, and $g_{(Z)} = 2.25(2)$. These results are in reasonable agreement with the parameters derived from the DC magnetic data (Supplementary Fig. 9d). It is clear that the simulation using a positive $D$ value is in good agreement with the experimental data. This result clearly indicates that this compound exhibits easy plane magnetic anisotropy. The EPR resonances at 1.49, 2.22, and 4.04 T at 120 GHz originate from energy transitions between the ground Kramers doublet with the field along the $Y$-, $X$-, and $Z$-directions, respectively (Supplementary Fig. 10).

**Angular dependence of the magnetization**. To measure the magnetic anisotropy and its changes, angular dependence measurements on the single crystal magnetic susceptibility at 5 and 190 K were performed[33,34]. The outcomes of isothermal rotations along the $x$-, $y$-, and $z$-directions are shown in Fig. 4d, where it can be seen that the susceptibility is highly dependent on the orientation of the measured single crystal. The minima values in the $z$-rotation are smaller than those of the $x$- and $y$-rotations, which indicates that this complex has an easy-plane-type magnetic anisotropy. No significant shifts in the maxima and minima for the same phase are observed with an increase in the temperature (Supplementary Fig. 11). From 5 to 190 K, both the values and positions of the peaks in the $x$-, $y$-, and $z$-rotations considerably move, which suggests that the magnetic anisotropy changes after structural transitions. The susceptibility tensors with respect to the experimental frame were determined by simultaneously fitting these curves (Supplementary Table 6). The directions of the hard axes for this complex are close to the Co1–O1w bond and the molecular plane (Fig. 4e and Supplementary Fig. 12). The hard axes form the angle of 49.8° with the crystallographic $a$-axis at 5 K and change to 61.4° at 190 K. The experimental hard axes are consistent with the ab initio calculated axes, with deviations of 21.1° at 5 K and 15.5° at 190 K. These deviations could be due to the vibronic coupling in the complex, temperature effect and error limits of experimental measurements and the water orientation from the XRD analysis[35–37]. Combined

with the Hamiltonian parameters $D$, $E$, $g_{(X)}$, $g_{(Y)}$ and $g_{(Z)}$ obtained from HF-EPR analysis, the temperature-dependent anisotropic magnetic susceptibility that is in good agreement with the experimental value can be reproduced (Supplementary Fig. 7e). The deviation of the magnetic susceptibility along the $y$- and $z$-axis is mainly derived from the assumption that $g_{(X)} = g_{(Y)}$. Here, the anisotropic susceptibility with apparent easy-axis nature is related to the non-collinearity of the magnetic principal axes ($X$-, $Y$-, and $Z$-axis) with the crystallographic directions ($x$-, $y$-, and $z$-axis). When we were calculating the anisotropic magnetic susceptibility of the crystal, we properly considered the relationship between the magnetic principal axes and the crystallographic axes and make the transformation.

**Effect of water orientation on magnetic anisotropy analyzed by ab initio calculations**. To gain insights into anisotropic changes in the magnetic susceptibility of the single crystal, the change in the $g$ tensor induced by structural transition was investigated through the ab initio analysis using the complete active space self-consistent field/N-electron perturbation theory (CASSCF/NEVPT2) approach. The calculated magnetic susceptibility for the microcrystal sample is in good agreement with the experimental data, especially for the LTp structure (Supplementary Fig. 7b). The principal components of the $g$ tensor $(S = 3/2)$ at different phases, listed in Supplementary Table 7, are highly anisotropic, i.e., $g_{(Z)} << g_{(X)} < g_{(Y)}$, which means that complex **1** exhibits easy plane magnetic anisotropy. The $g$ tensor values for the LTp are consistent with the results of the HF-EPR experiments. These results show the temperature dependence of the magnetic susceptibilities $\chi_{\mathrm{M}x}T$, which is in good agreement with the experimental data (Fig. 4f and Supplementary Fig. 7c, the $\chi_{\mathrm{M}x}T$ curve is mainly discussed here because it is measured along the crystallographic $a$-axis in HTp and LTp and shows an obvious two-step transition). The structural transformation of complex **1** in response to the temperature led to a remarkably large shift in the direction of the $g$ tensor. According to the ab initio calculations, the hard axis of the $g$ tensor deviates from the $a$-axis ($x$-direction for the magnetic measurements) in the HTp structure by $\Phi_{Za} \approx 57.9°$ and in the LTp structure by $\Phi_{Za} \approx 45.2°$. However, the principal $g$ values only exhibit minor changes (within 3%) on the phase transition. As a consequence, the $g$ value in the $x$-direction of the single crystal decreases from 2.44 for the HTp structure to 2.23 for the LTp structure. This directional change in the $g$ tensor corresponds to the large decrease (23%) observed in the experimental $\chi_{\mathrm{M}x}T$ value. The directional shift of the $g$ tensor affects the $g$ value in the $yz$ plane of the single crystal, and the corresponding $\chi_{\mathrm{M}y}T$ and $\chi_{\mathrm{M}z}T$ values show clear changes during phase transition. The results of the calculations are in good agreement with the experimental data (Supplementary Fig. 7d). This suggests that the atypical two-step magnetic anisotropy transition of the single crystal of **1** is induced by the substantial shift in the direction of the molecular magnetic anisotropy that arises owing to the changes in the molecular structures.

The switching of the magnetic anisotropy and the change in the orientation of the water molecule suggest that the rotation of the water molecule at the Co(II) center plays a pivotal role in determining the unexpected switching of the hard axis. To investigate the effect of the water rotation, ab initio calculations were performed on a model (model 1), where the water molecule in the LTp structure was rotated around the Co1–O1w axis in the range from −30° to 30° at intervals of 5° while keeping other ligand positions fixed (Fig. 5a; the positive direction of rotation is defined as the operation that makes the virtual structure closer to that of the HTp structure). The results of the calculations show that the values and orientations of the principal components of

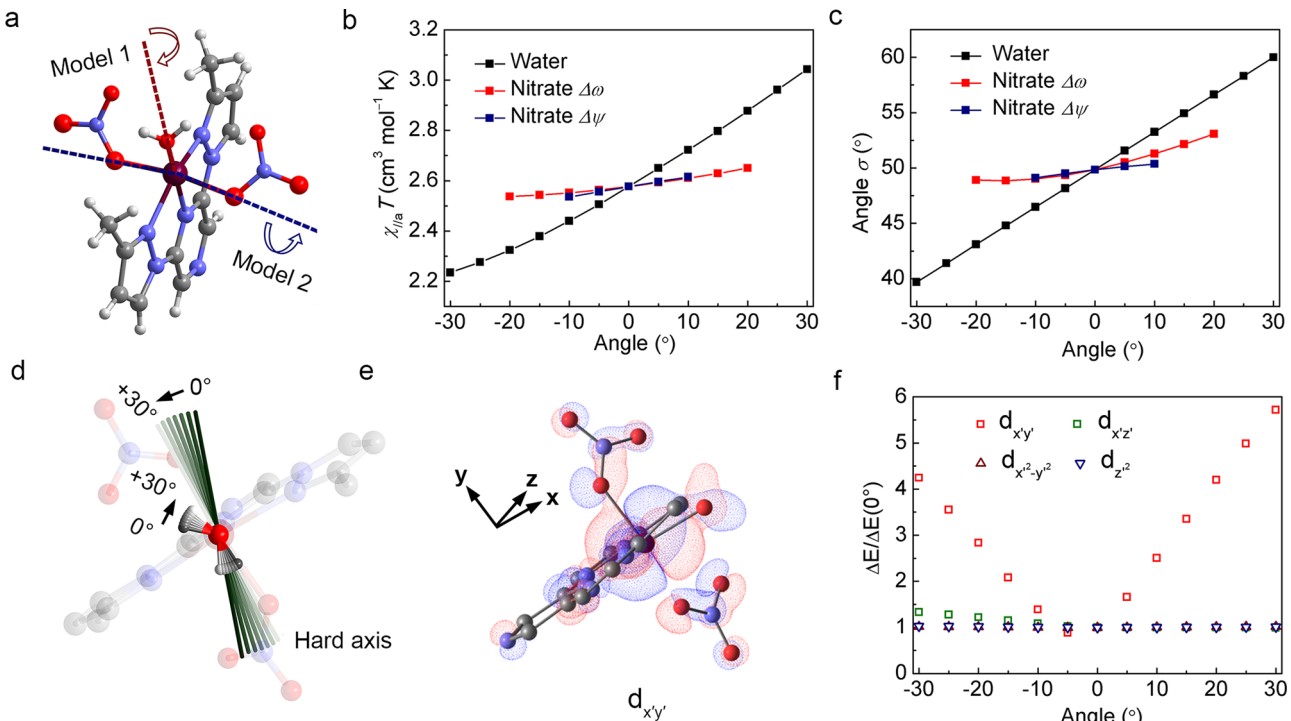

**Fig. 5 Variations in the magnetic anisotropy parameters with the rotation of water and nitrates from ab initio calculations. a** Two molecular models (model 1, rotation of the water molecule; model 2, rotation of nitrates) based on the structure of LTp for calculations. **b** Magnetic susceptibility along the $a$-axis. **c** Deviation in the angle, $\sigma$, change with the rotation angle in the two models. **d** Schematic diagram of the change of the direction of the magnetic hard axis with the rotation of the water molecules. **e** Overlap between the $d$ orbital centered on the cobalt and the $p$ orbitals of the oxygen from the water and the nitrates in the highest fully occupied $d$ orbital ($d_{x'y'}$), where the red color corresponds to the regions where the phase of the wave function is positive, and the blue color corresponds to the regions where the phase of the wave function is negative. The reference coordinate axes are defined in Supplementary Fig. 14a. **f** Energy differences between the $d$ orbitals and the $d_{y'z'}$ orbital ($\Delta E$) with respect to those of the initial state [$\Delta E(0°)$] change with the rotation of water. Maroon, Co; gray, C; blue, N; red, O; light gray, H.

the $g$ tensor considerably change with the rotation of the water molecule (Fig. 5c, d and Supplementary Fig. 13a). As shown in Fig. 5b, the rotation of the water molecule by 10° leads to a change in the magnetic susceptibility along the $a$-axis of the magnitude of about 5.6%. This result indicates that the orientation of the water molecule greatly contributes toward the magnetic susceptibility along the $a$-axis at 300 K. The changes in the average magnetic susceptibility were observed to be considerably small, which is in good agreement with the experimental result. The effective $g$ tensor values slightly change with positive rotations, whereas the direction of the $g$ tensor considerably varies. The angle ($\sigma$) between the hard axis and the $a$-axis changes from 49.8° to 60.0° after the water rotates by 30°. This result indicates that the changes in the magnetic susceptibility along the $a$-axis can be mainly attributed to the rotation of the water molecule. To support this result, ab initio calculations were performed on another model (model 2), where the dihedral angle $\varphi$ is the same as that in the LTp structure, whereas the dihedral angles $\omega$ and $\psi$ are related to the coordination sphere of the Co(II) ion rotated in-plane around the Co1–O4 and Co1–O1 bonds (Fig. 5a). As shown in Fig. 5b, the rotation of the nitrate group by 10° leads to changes in the magnetic susceptibility along the $a$-axis of the magnitude of about 1.5%, which demonstrates that the orientation of the water contributes more toward the magnetic susceptibility than the effect of the rotation of both nitrates. When the same calculations were performed in the HTp structure, the same trend was observed for the above parameters (Supplementary Fig. 13b).

The water molecule coordinates to Co(II) through the strong $\sigma$ bond. However, the rotation of the water molecule has a lesser

influence on the nature of the $\sigma$ bond, which suggests that the interaction between $p\pi$ in the water molecule and $d\pi$ in Co(II) is essential for the anisotropy switching. Indeed, the ligand-field analysis based on the ab initio calculations reveals that the $\pi^*$ anti-bonding interaction between the $d$ orbitals centered on Co1 and O1w from the water molecule brings about considerable variations in the energy level of the highest fully occupied $d$ orbital (Fig. 5f and Supplementary Fig. 14a), which contributes the most to the configuration interaction and magnetic properties. This result is attributed to the overlap between the $d$ orbitals centered on Co1 and the $p$ orbital of O1w, which approaches the $d$-centered orbital in the direction favored by $\pi$ symmetry, whereas the overlap between the $d$ orbitals centered on Co1 and the $p$ orbital of O1(O4) from the nitrates has a lower $\pi$ character because of structural distortion (Fig. 5e and Supplementary Fig. 14b). Therefore, the orbital energy of the $d$ orbital and hence the directional magnetic susceptibility is more sensitive to the water rotation. In addition, the shape of the $d$ orbitals in this case will also change with the rotation of the water from theoretical calculations (Supplementary Fig. 15), which will help us to better understand the previous results. These explanations are in good agreement with the results of quantum calculations on the Dy(III) complex performed by Sessoli and co-workers[12,13]. In the Dy(III) complex, the calculated easy axis of magnetization also rotates when the apical water molecule is rotated, which was further explained through an in-depth theoretical analysis to be because of the covalent contribution of the water molecule. This work experimentally confirms the impact of the orientation of the water ligand in the complex on its magnetic anisotropy properties in the Co(II) complex, in which the bonding is more covalent in

character than in lanthanide complexes because the $4f$ valence orbitals are not really accessible.

To conclude, an exciting two-step water reorientation and the concomitant magnetic anisotropic change were observed in a cobalt(II) complex. The water molecule changed its orientation through rotation around the Co–O bond after the phase transition. Structural analysis, angular-resolved magnetometry studies, and theoretical calculations indicate that the rotational reorientation of the water molecule, which involves a $\pi^*$ anti-bonding interaction, is a key factor determining the magnetic anisotropy of the complex. This study highlights that the reversible rotational reorientation of coordinated water in response to external stimuli, such as temperature, can be realized in crystals, and such subtle structural changes of the water ligand can have drastic effects on the magnetic anisotropy. Consequently, the orientation of coordinated water will be definitely considered when analyzing the performance of materials or devices in the presence of water molecules. We believe that the discovery of the effect of the orientation of water ligands will be useful for improving the physical properties of materials and for further promoting the application of functional complexes, especially molecular magnetic materials, in devices and the bioimaging.

## Methods

All the reagents were obtained from commercial suppliers and were used without further purification, except for ligands 2,6-bis(3-methylpyrazol-1-yl)pyrazine (mprpz) and deuterated mprpz (mprpz-$d_{12}$) (Supplementary Methods).

**Synthesis of complex 1 and deuterated complex 1**. The target complex [Co(ONO)$_2$(H$_2$O)(mprpz)] (**1**) was prepared by mixing an acetone solution of mprpz (0.01 M, 10 cm$^3$) with an acetone solution of Co(NO$_3$)$_2$·6H$_2$O (0.01 M, 10 cm$^3$) in a test tube. Diethyl ether vapor was slowly evaporated into the resulting pink solution in a sealed tube. Purple crystals appeared after a few days, with a yield of 67%. Large crystals suitable for magnetization measurements were synthesized by slowly evaporating the pink solution; analyzed (calculated) (%) for C$_{12}$H$_{14}$CoN$_8$O$_7$: C, 32.63 (32.74); H, 3.17 (3.22); N, 25.40 (25.47). The same method of preparation was used for the deuterated complexes [Co(ONO)$_2$(D$_2$O)(mprpz)] (**1**-$d_2$) and [Co(ONO)$_2$(D$_2$O)(mprpz-$d_{12}$)] (**1**-$d_{14}$) using acetone-$d_6$ solvent, mprpz-$d_{12}$ and Co(NO$_3$)$_2$·6D$_2$O (The Co(NO$_3$)$_2$·6D$_2$O was prepared by the repeated recrystallization of Co(NO$_3$)$_2$·6H$_2$O with D$_2$O).

**Single-crystal X-ray diffraction (XRD)**. XRD measurements of complex **1** at 70 and 140 K (LTp and Ip) were performed on a Rigaku Saturn CCD area detector with graphite monochromated Mo Kα radiation ($\lambda = 0.71075$ Å). The measurements for complex **1** at 190 K (HTp) were performed using synchrotron radiation at beamline BL02B1 of SPring-8 (Hyogo, Japan) with shorter wavelengths ($\lambda = 0.41470$ Å) using a Rigaku Mercury II detector. The temperature-dependent structure of **1**-$d_{14}$ was also determined on a Rigaku Saturn CCD area detector with graphite monochromated Mo Kα radiation ($\lambda = 0.71075$ Å). The high-resolution single-crystal XRD analysis for complex **1** at 190 K (1_190K_hr), 140 K (1_140K_hr), and 70 K (1_70K_hr) were performed using synchrotron radiation at beamline BL02B1 of SPring-8 (Hyogo, Japan) with shorter wavelengths ($\lambda = 0.41195$ Å) using a Rigaku Mercury II detector. The crystal was enveloped in a temperature-controlled stream of dry nitrogen gas (a mixture of nitrogen and helium gas) during data collection. The structures were solved by direct methods and were refined via full-matrix least-squares on $F^2$ using the SHELX software program[38] with anisotropic thermal parameters for all non-hydrogen atoms. All the H-atoms were located using a difference Fourier map and refined freely. The configuration of the methyl groups in 1_190K_hr was refined with restraint DFIX. The X-ray crystallographic coordinates for the structures reported in this study have been deposited under deposition numbers CCDC 1991940-1991942, 2058027-2058029, and 2065093-2065095 (http://www.ccdc.cam.ac.uk/data_request/cif).

**Crystal structure of 1-$d_{14}$ and 1 determined by powder neutron diffraction**. Neutron powder diffraction patterns were recorded using a time-of-flight powder Super-HRPD (BL08) diffractometer located at the Spallation Neutron Source at the Japan Proton Accelerator Research Complex (J-PARC). Powder samples weighing 1.7003 and 1.4027 g (**1**-$d_{14}$ and **1**) were charged into vanadium cells (3 mm in diameter and 65 mm in length) to mount onto the diffractometer. Patterns were collected at 71 and 194 K for **1**-$d_{14}$ and at 70 and 193 K for **1**. The crystal structures of **1**-$d_{14}$ were refined by the Rietveld method using Z-Rietvelds[39]. The results of the refinements are provided in cif files (CCDC 2020263-2020264).

**Angular dependence of the magnetization**. The indexed single-crystal of complex **1** was glued onto a Teflon square, and the ensemble was mounted on a goniometric head to conduct angular-resolved magnetometric measurements. The $x$, $y$, and $z$ orthogonal directions, parallel to the rotation axes, have the following composition in the crystallographic cell in terms of $a$, $b$, and $c$ at 190 and 70 K; $x = a$; $y = 0.3697\,a + 0.9291\,b$; $z = 0.1031\,a + 0.01322\,b + 0.9946\,c$ (190 K); $x = a$; $y = 0.3884\,a + 0.9215\,b$; $z = 0.1202\,a – 0.01341\,b + 0.9927\,c$ (70 K). Three orthogonal rotations were then performed to extract the angular dependence of the magnetic susceptibility. The data were then fitted using the equation: $\chi^{\mathrm{rotz}}(\theta) = \chi_{xx}\cos^2\theta + \chi_{yy}\sin^2\theta + 2\,\chi_{xy}\cos\theta\,\sin\theta$, where $\theta$ is the angle formed by the magnetic field and the $x$-axis.

**Measurements**. DC and AC magnetic susceptibility measurements were performed on an MPMS-5S SQUID magnetometer. For the measurements, single crystals and polycrystalline samples were fixed with grease and hexadecane, respectively, to prevent crystallite torqueing. All data were corrected for diamagnetic contributions for both the hexadecane and the individual samples. HF-EPR measurements were performed on a locally developed spectrometer at the Wuhan National High Magnetic Field Center, People's Republic of China, using a pulsed magnetic field[40–42]. The tunable microwave frequencies were generated by a combination of Gunn (Millitech) and backward wave oscillators (Institute of General Physics, Moscow, Russian Federation). The data were recorded using an InSb hot-electron bolometer (QMC Ltd., Cardiff, U.K.). Samples were ground with KBr and were pressed into pellets to reduce the field-induced torqueing effect. Heat capacity measurements on the polycrystalline sample were performed between 8 and 300 K using a laboratory-made low-temperature adiabatic calorimeter[43]. After buoyancy correction, 0.11813 g of the sample was loaded into a gold-plated copper cell filled with helium gas, which functions as a heat-exchange medium, and sealed together at ambient pressure using an indium gasket. Thermometry was performed with a rhodium–iron alloy resistance thermometer (nominal 27 Ω, Oxford Instruments), which was calibrated on the basis of the international temperature scale of 1990 (ITS-90). DSC measurements were performed on a Seiko EXSTAR 6000 instrument using cooling and heating rates of 10 K min$^{-1}$. Powder XRD analysis was performed on sample plates at room temperature at 50 kV and 300 mA using a Cu target (RIGAKU TTR-III). IR spectra were recorded on a JASCO FT/IR-600 Plus spectrometer in the 400–4000 cm$^{-1}$ region. Samples were ground with KBr and then pressed into disks. Elemental analyses were conducted using a Vario EL elemental analyzer. Proton nuclear magnetic resonance ($^1$H-NMR) spectra of the synthesized compounds were measured with AVANCE III HD 400 MHz (Bruker Co. Ltd., Massachusetts, USA).

**Computational methods**. Ab initio calculations were performed using the ORCA 4.0.0 package[44,45], adopting the CASSCF/NEVPT2 approach[46,47]. The scalar relativistic effect was considered by the zero-order regular approximation. Spin–orbit coupling was calculated using a mean-field spin–orbit coupling operator[48]. The mixing of CI eigenfunctions was calculated using the quasi-degenerate perturbation theory. Magnetic susceptibility tensors were calculated and rotated to the measurement coordinate frame to compare with the experimental results. The segmented all-electron relativistically contracted Ahlrichs basis sets were used for all atoms (def2-TZVPP for Co atom and def2-TZVP for other atoms)[49,50]. The coordinates of all non-hydrogen atoms were adopted from experimental XRD measurements, whereas the positions of the hydrogen atoms were optimized with the inclusion of two guest nitric acid at the B3LYP/def2-TZVP level[49,51–53] with Grimme's empirical dispersion correction[54,55] (Supplementary Data 1). When magnetic properties were calculated, the two nitric acid molecules were removed from the structures. The active space comprises the seven electrons of the Co(II) ion in the five $d$ orbitals (CAS(7,5)). All the 15 quartets and 40 doublets were considered by state-average CASSCF. Finally, we used a more robust active space (CAS(17,10)) containing all the five metal–ligand bonding orbitals and five 3$d$-centered orbitals with all the 17 electrons on them and the results got improved. The ligand-field orbital energies and ligand-field parameters were extracted from the ab initio ligand field theory (AILFT) module implemented in the ORCA package based on CAS(7,5) calculations for all model systems[56].

## Data availability

The X-ray crystallographic data for the structures at different temperatures reported in this study have been deposited at the Cambridge Crystallographic Data Centre (CCDC), under deposition numbers CCDC 1991940-1991942, 2058027-2058029, 2065093-2065095 and 2020263-2020264. These data can be obtained free of charge via http://www.ccdc.cam.ac.uk/conts/retrieving.html or from CCDC (12 Union Road, Cambridge CB2 1EZ, UK; Fax: +44 1223 336033; e-mail: deposit@ccdc.cam.ac.uk). Additional data are available from the authors upon request.

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

## Acknowledgements

We thank Prof. Masaaki Ohba for his help in angular-resolved magnetometry studies, Prof. Kazunari Yoshizawa and Prof. Yoshihito Shiota for their help in theoretical calculations. This work was supported by JSPS KAKENHI Grant Number 18K14244 and 20H00385. This work was supported by the MEXT Project of "Integrated Research Consortium on Chemical Sciences". The synchrotron radiation experiments were performed at the BL02B1 of SPring-8 with the approval of the Japan Synchrotron Radiation Research Institute (JASRI) (Proposal No.2018A1213). Neutron powder-diffraction were at BL08 located at the Spallation Neutron Source at the Japan Proton Accelerator Research Complex (J-PARC) (Proposal No.2019A0226). This work was partly supported by Nanotechnology Platform Program (Molecule and Material Synthesis) (JPMXP09S20MS1076) of the Ministry of Education, Culture, Sports, Science and Technology (MEXT), Japan.

## Author contributions

S.S. and O.S. designed the study, conducted experiments, and wrote most of the paper. S.W. performed the calculations. M.H., P.M., Z.T., S.T., and T.K. performed the neutron powder diffraction measurements. Y.M. and M.N. performed the heat capacity measurements. T.X., Z.W., and Z.O. performed HF-EPR measurements and fitted the data. T.N., J.L., and S.K. assisted in structural measurements and results analyses. All authors discussed the results and commented on the manuscript.

## Competing interests

The authors declare no competing interests.
