## [Peer Review File · Nature Communications]

REVIEWER COMMENTS

Reviewer #1 (Remarks to the Author):

The presented paper focus on the importance of the water mobility in determining the magnetic properties of Co(II) complexes following the idea recently reported and only computationally proved on the effect of the water molecule rotation on the magnetic anisotropy easy axis direction of the DyDOTA complex (Chem. Sci.10, 7233–7245 (2019)). Differently, they proved experimental evidences other then computational results. Indeed, they claim to an exciting two-step water reorientation and the concomitant magnetic anisotropic change in the cobalt(II) complex. The water molecule changed its orientation through rotation around the Co–O bond after the phase transition.

To prove it, structural analysis, angular-resolved magnetometry studies, and theoretical calculations are presented. π^* anti-bonding interaction, is claimed as a key factor determining the magnetic anisotropy of the complex. The most appealing aspects of the paper are that the rotational reorientation of coordinated water can be controlled in crystals. Consequently and that the orientation of coordinated water will be definitely have to be considered when analyzing the performance of materials or devices in the presence of water molecules.

Such a paper seems to give a strong support to what hinted in the parent Chem. Sci. paper pointing out that the effect of the orientation of ligand waters can be another useful degree to take into consideration for the interpretation of the physical properties of materials and to improve them in different field from spintronics or biochemical systems.

The paper is well written and a high readability and it can potentially be of interest for the publication in Nature Communications

However, I have the following concerns:

- Since the structural changes are tiny it would have been better to perform an optimization including more than one molecules or, better, perform a periodic calculations since x-ray data are present for the different temperatures. Indeed, since the orientation of the water molecule is given by two nitrate groups belonging to different molecules, they should be included and not just keeping them fixed as done in the paper.
- The authors report that “To consider the ligand field effect more completely, two ligand orbitals and the 4d double shell were also included in the active space (CAS(11,12)). “ First of all, they do not give any information about the nature of ligand orbitals added to the CAS. However, the main concerns are related to the fact that the addition of only two ligand orbitals in the CAS can be likely not consistent if the whole coordination sphere is not completely considered. Moreover, the quartets should be 10 and not 15.
- Effects of water/nitrates rotations on easy magnetization axis should be also presented on the HS spin state.
- Related to the points above, the computed large deviations observed for the computed and the experimental magnetic easy axis orientations are, in my opinion, too large.
- The complexes interact one each other through pi systems. The authors have not

considered the possible exchange interactions that can take place even in the experimental data. They should take it into consideration after having computed it at the ab initio level by a DFT Broken Symmetry approach, too.

- Fig 5: A reference system to understand how the M-L bonds are oriented should be provided as the energy variation of the d_{yz} orbital. The prime symbols should be explained.

On the basis of my concerns reported above I could recommend the publication in Nature Communications if all the above points will be accounted for and their replies fully justified.

Reviewer #2 (Remarks to the Author):

The manuscript by Sato et. al. describes a high-spin coordination complex of Co(II) with a coordinated water molecule, two coordinated nitrate ions, and a tridentate nitrogen donor ligand. In the solid state, this complex undergoes two structural phase transitions as function of temperature at ca. 113K and 157K. Through the intramolecular hydrogen bonding between the coordinated nitrate ions and the water molecule, these structural phase transitions bring about a step wise rotation of the coordinated water molecule, relative to the remaining coordination sphere, around the water-cobalt bond direction. The pi-bonding anisotropy of the coordinated water molecule leads to a concomitant change in the orientation of the magnetic susceptibility tensor.

The importance and novelty of this study is, according to the authors, deriving from it a) being the first experimental example of the importance of the anisotropic metal-water interaction for magnetic properties b) that the "reorientation of coordinated water can be controlled in crystals, ..." and "the orientation of coordinated water will be definitely considered when analyzing the performance of materials or devices in the presence of water molecules."

This is a very nice study, it is well conducted, quite encompassing, and clearly reported. I think it is complete and suited for publication. I am, however, somewhat hesitant to recommend it for Nature Comms., mainly because I think that the impact and scope of appeal are not quite matching the high requirements of this journal. There are two primary arguments relating directly to how the importance of the study is presented in the manuscript. One concerns novelty and one concerns the impact in terms of providing an experimental handle on the properties of "materials or devices".

Addressing the novelty issue first:

I think that the claim that "Until now only theoretical studies have been conducted" (line 87), is incorrect. Several studies on water complexes of transition metal ions discuss the importance of the anisotropic donor properties of water and the interplay with the electronic structure of the metal center. This has led to classification of alums with classes exhibiting different water coordination modes (cf. e.g. Beattie, et.al, Dalton Trans, 1981, 2105 as well as Armstrong et. al., Dalton trans, 1983, 1973). Notably, the direct correlation between water coordination geometry and spectroscopic as well as magnetic properties have been analyzed in a series of papers by P.L.W. Tregenna-Piggott. Two examples: "Influence of the Mode of Water Coordination on the Electronic Structure of the $[V(OH_2)_6]^{3+}$ Cation", Journal of Solid State Chemistry 1992, 145, 460. and "Structure and Bonding of the Vanadium(III) Hexa-Aqua Cation. 1. Experimental Characterization and Ligand-Field Analysis" Inorg. Chem. 2004, 43, 8049. In the latter of these papers the influence of water coordination geometry on magnetic anisotropy is explicitly discussed. I think this deserves to be referenced, but also that it constitutes direct "prior art".

Concerning the consideration of the water coordination geometry in the context of materials and devices:

It is true that water - like many other ligands - have inherently anisotropic interactions with metal ions. Arguably, other pi-anisotropic ligands are more amenable than water to provide conformational control. Quantifying pi-anisotropies is indeed a very important step forward in understanding and analyzing electronic structure and magnetic properties. However, the quantum leap of controlling properties through conformational control lies well beyond the observation of phase transitions with associated conformational changes. Without the design strategy (cf. the statement "This study highlights that the rotational reorientation of coordinated water can be controlled in crystals") for generating the phase transitions, (cf. the statement "This study highlights that the rotational reorientation of coordinated water can be controlled in crystals", line 313) this remains, in my opinion, an interesting observation, but not control.

Reviewer #3 (Remarks to the Author):

The manuscript "Water-oriented magnetic anisotropy transition" by Sheng-qun Su et al. describes a multi-technique study of a Co(II) complex. The authors are trying to ascribe the change of magnetic anisotropy of the Co(II) center ion to the rotation of the water ligand.

While in my opinion the findings are interesting and the experimental results seem to be of good quality I have major reservations from the magnetism point of view. The authors claim an easy-plane type magnetic anisotropy with a positive D value in the order of 30 cm⁻¹. This is not consistent with the magnetic susceptibility data that they show in Figure 4b,d,f. Also, I am not convinced that the positive D value can be extracted with high certainty only from low-temperature HF-EPR measurements. With this D value, the higher lying $m_s = \pm 3/2$ states will not be populated, and a crossing with the $\pm 1/2$ states is not expected in experimental magnetic fields rendering the zfs inaccessible to HF-EPR at this low temperature.

Another point: The authors claim that they experimentally determine the water molecule rotation as a reason for the change of magnetic anisotropy, going beyond the case described in [G. Cucinotta et al., *Angew. Chem. Int. Ed.* 51, 1606 (2012)]. Concept-wise, calculations are still needed here to identify the water molecule rotation as a reason for the anisotropy change, among many other structural changes that occur during the phase transitions, which renders the present study less original than it is claimed by the authors.

Other points:

- 1- The spin Hamiltonian model used in the work should be clearly noted down, e.g., in the SI.
- 2- The authors should demonstrate that the spin-Hamiltonian model (D,E,g_x,g_y,g_z parameters) can reproduce the directional dependence along {x,y,z} of the χ^*T product shown in Fig 4b.
- 3- The assignment of the HF-EPR transitions is not consistent between Figs. 4c and S10
- 4- Calculations: The authors should show the outcome of the ab-initio calculations for the χ^*T of the single crystal (directional dependence along {x,y,z}, analogous to the experimental data shown in Fig 4).

5- The effective magnetic moment (μ_{eff}) is calculated to be 4.934 μ_B (page 5, Section "Anisotropic magnetic susceptibility transition. "). The authors should explain how such a large value is compatible with a $S=3/2$ system.

6- I am not able to judge seriously the structure determination from XRD/NPD.

Reviewer #4 (Remarks to the Author):

This manuscript reports on the experimental and theoretical results of magnetic anisotropy influenced by water rotation in the Co-coordinating compound. The claimed experimental observation of water rotation and its effect on the anisotropy is significant. However, I don't think the claim on the structures is convincing enough at the current shape. In general, H atom could not be refined reliably by SHELXL (which is applied in this work) based on single-crystal X-ray diffraction data, especially for the reported data, all three datasets have a rather humble resolution (ca. 0.77 Å), considering the applied Mo radiation ($\lambda = 0.71075$ Å) at 70 and 140 K, and the Spring-8 high-resolution beamline with $\lambda = 0.41470$ Å at 190 K, much better resolution of the datasets should be obtainable. Probably, the authors cut off the high-resolution reflections, which is vital to the high accuracy measurement of the electron density, so critical to the determination of the H atoms in this case. To alleviate the doubts on H determination, the authors tried neutron diffraction, however, with powder sample. I don't think powder neutron diffraction would give more reliable results in this case. Single-crystal neutron diffraction would be perfect here to determine the H and may corroborate the claims made by the authors. The authors are strongly recommended to do this. However, considering the difficulties (probably not so in the research environment of the authors) to conduct single-crystal neutron diffraction, very high-resolution X-ray diffraction would be an alternative.

Except for the serious issue mentioned above, the other concerns are listed in the following: 1, the dihedral angle ϕ (defined between the plane defined by the N1, N3, and N5 atoms and the plane of the coordinated water molecule) is used to evaluate the water molecule rotation around the Co–O bond, please keep in mind, the Co–O bond is not on the defined N1, N3, and N5 plane.

2, the supplementary crystal data of "1-d14" is not supplied. The data parameters presented in Supplementary Table 1 indicates that the structure at 120 K is already the low-temperature phase of the proton cousin, as discussed in the manuscript, the deuterated compound shows a 3 K increase of phase transition, then it should be still at the intermediate phase at 120 K, similar to the 150 K, why not?

3, the data at 70 K is a little strange to me since the low-temperature X-ray diffraction has lower thermal vibration, normally the data quality should be relatively higher than high-temperature data, at least not worse, it is worse than the 140 and 190 K datasets in this case. I recommend double-checking the data (especially for space group determination). For the missed data of "1-d14", similar issues present for 120 and 150 K datasets. I can imagine one possibility: the phase transition is not completed. If this is the case, annealing during cooling down may help.

4, the refinement of the H atom is not consistent (70 and 140 K: several H atoms were added as riding model, even for the critical water H atoms, the ADPs are refined as riding model; 190 K, several H atoms were added as riding model) for the three datasets, which is critical for the comparison. I recommend refining all the H atoms freely just as the authors did for the water H atoms at 190 K.

5, the discussion on the phase transition from the structural perspective looks not so clear. I feel like the four nitrate ions nearby the water ligand are competing to generate H-bond with

the water molecule.

6, the π - π interaction is not so clear, it looks like there is no real π - π interaction in the crystal lattice, since even the highlighted π - π interaction in Figure 1d doesn't show overlap on the π electron density, I would prefer to call it C-H- π interaction.

7, it looks like indexing of the face roughly perpendicular to z-direction as 100 is a mistake in Figure 4a.

REVIEWER COMMENTS

Reviewer #1 (Remarks to the Author):

The presented paper focus on the importance of the water mobility in determining the magnetic properties of Co(II) complexes following the idea recently reported and only computationally proved on the effect of the water molecule rotation on the magnetic anisotropy easy axis direction of the DyDOTA complex (Chem. Sci.10, 7233–7245 (2019)). Differently, they proved experimental evidences other then computational results. Indeed, they claim to an exciting two-step water reorientation and the concomitant magnetic anisotropic change in the cobalt(II) complex. The water molecule changed its orientation through rotation around the Co–O bond after the phase transition.

To prove it, structural analysis, angular-resolved magnetometry studies, and theoretical calculations are presented. π^* anti-bonding interaction, is claimed as a key factor determining the magnetic anisotropy of the complex. The most appealing aspects of the paper are that the rotational reorientation of coordinated water can be controlled in crystals. Consequently and that the orientation of coordinated water will be definitely have to be considered when analyzing the performance of materials or devices in the presence of water molecules.

Such a paper seems to give a strong support to what hinted in the parent Chem. Sci. paper pointing out that the effect of the orientation of ligand waters can be another useful degree to take into consideration for the interpretation of the physical properties of materials and to improve them in different field from spintronics or biochemical systems.

The paper is well written and a high readability and it can potentially be of interest for the publication in Nature Communications

However, I have the following concerns:

- Since the structural changes are tiny it would have been better to perform an optimization including more than one molecules or, better, perform a periodic calculations since x-ray data are present for the different temperatures. Indeed, since the orientation of the water molecule is given by two nitrate groups belonging to different molecules, they should be included and not just keeping them fixed as done in the paper.

Reply: Thank you for your suggestions. Unfortunately, the periodic calculations are quite time-consuming. Here we adopted two optimization models to take the intermolecular hydrogen bonds into account: (1) a hydrogen-bonded molecular trimer and (2) a molecule with two guest nitric acids. The calculations were performed using the B3LYP functional with Grimme's empirical dispersion correction based on the LT X-ray structure, whose properties were fully characterized. Indeed, differences in the hydrogen positions were found. However, because of the very small energy difference of these optimized structures in different water orientations, these results may suffer from significant dependence on functional and basis sets. To evaluate the validity of these models, CASSCF(7,5)/NEVPT2 calculations were also performed on these two newly optimized geometries and were compared with the former results. The obtained magnetic parameters are listed in Supplementary Table 6 and were compared with the experimental effective g values (which required the least fitting parameters and restrictions). It

can be found that neither model provides a better match with the experimental results compared with our model as described in the original manuscript. These results suggest that the original model is reasonable for accounting for the observed phenomenon and can provide a correct semi-quantitative description of it. We hope that you find our explanation to be satisfactory.

Supplementary Table 6 Principal components of the g tensor and the zero-field splitting parameters D and E .

	$(S = 3/2)$			$(S = 1/2)$			D (cm ⁻¹)	E (cm ⁻¹)
	g_x	g_y	g_z	g_x	g_y	g_z		
Exp. (HF-EPR) [*]	2.45	2.45	2.25	3.879	5.795	2.191	32.0	4.3
Calcd. (LTp) ^a	2.414	2.625	2.077	3.603	6.316	2.080	50.2	8.2
Calcd. (LTp) ^e	2.437	2.647	2.083	3.727	6.277	2.138	53.4	8.1
Calcd. (LTp) ^f	2.415	2.627	2.080	3.598	6.319	2.080	49.9	8.2
Calcd. (LTp) ^g	2.400	2.632	2.084	3.378	6.490	2.021	48.4	9.2

^{*} Simultaneous fitting of the HF-EPR and magnetic susceptibility. g_x and g_y are restricted to be the same.

^a Calculation based on the H-optimized geometry and active space constructed from the five 3d orbitals.

^e Calculation based on the H-optimized geometry and active space constructed from the five 3d orbitals and their bonding partner orbitals.

^f Calculation based on the H-optimized geometry, including two nitric acids forming H-bonds with a water molecule and active space constructed from the five 3d orbitals.

^g Calculation based on the H-optimized geometry, including two neighboring complex motifs forming H-bonds with a water molecule and active space constructed from the five 3d orbitals.

- The authors report that “To consider the ligand field effect more completely, two ligand orbitals and the 4d double shell were also included in the active space (CAS(11,12)).” First of all, they do not give any information about the nature of ligand orbitals added to the CAS. However, the main concerns are related to the fact that the addition of only two ligand orbitals in the CAS can be likely not consistent if the whole coordination sphere is not completely considered. Moreover, the quartets should be 10 and not 15.

Reply: Thank you for your insightful comments. The two ligand orbitals mentioned in the original manuscript are the two highest 3d- π bonding orbitals. Indeed, it is unbalanced. In the revised manuscript, we use a more robust active space (CAS(17,10)) containing five 3d-centered metal-ligand anti-bonding orbitals and their five metal-ligand bonding partner orbitals with all the 17 electrons on them. The result was improved and we updated the corresponding figures in the main text.

Regarding the number of quartet states, considering that some ligand-centered orbitals have been included in the active space, we slightly increased the number of calculated configurations in the state-averaged CASSCF to check whether there is low-energy ligand-to-metal charge transfer states or not. We carefully checked our results and found no sign of these. All the configurations from different occupation numbers on the d orbitals have been considered with the original “10 quartets + 40 doublets” scheme. Thus, in the latest calculations, we used 10 quartets and 40 doublets as suggested. We hope that you are satisfied with these corrections.

- Effects of water/nitrates rotations on easy magnetization axis should be also presented on the HS spin state.

Reply: Thank you for your suggestions. The same calculations to evaluate the effects of water/nitrates rotation on the magnetic properties were performed using the HTp structure (Supplementary Fig. 13b). The same trend was observed for the changes of magnetic susceptibility along the a-axis, average magnetic susceptibility, effect of the g tensor values ($S = 1/2$), angle σ and magnetic hard axis according to the rotation angle of the water/nitrates.

- Related to the points above, the computed large deviations observed for the computed and the experimental magnetic easy axis orientations are, in my opinion, too large.

Reply: Thank you for your comments. Admittedly, significant deviations can be found in the orientation of the calculated and experimental principal axes. However, most such comparisons have been made using lanthanide single-ion magnets, which are dominantly determined by electrostatic interactions. Calculations for magnetic properties of transition-metal complexes with competing spin-orbit coupling and ligand-field splitting like the present compound are challenging. One possible reason for the calculated orientation, especially for the HT phase, could be the vibronic coupling; the thermal ellipsoid of the water O atom is significantly distorted, suggesting that the water molecule possibly suffers from severe thermal vibrations. Considering the important role of water as shown in our study, it is not surprising that the vibronic interaction will affect the magnetic properties. Of note, because of error limits of the experimental measurements, the water orientation and temperature effect from the XRD analysis will also contribute to this discrepancy³⁵⁻³⁷.

35 Atanasov, M., Zadrozny, J. M., Long, J. R. & Neese, F. A theoretical analysis of chemical bonding, vibronic coupling, and magnetic anisotropy in linear iron(II) complexes with single-molecule magnet behavior. *Chem. Sci.* **4**, 139–156 (2013).

36 Rigamonti, L. *et al.* A Pseudo-Octahedral Cobalt(II) Complex with Bispyrazolylpyridine Ligands Acting as a Zero-Field Single-Molecule Magnet with Easy Axis Anisotropy. *Chem. Eur. J.* **24**, 8857–8868 (2018).

37 Qian, K. *et al.* Does the thermal evolution of molecular structures critically affect the magnetic anisotropy? *Chem. Sci.* **6**, 4587–4593 (2015).

- The complexes interact one each other through pi systems. The authors have not considered the possible exchange interactions that can take place even in the experimental data. They should take it into consideration after having computed it at the ab initio level by a DFT Broken Symmetry approach, too.

Reply: Thank you for your suggestions. The BS-DFT calculations based on a dimer model show that the estimated value of the intermolecular magnetic coupling constant is -0.01 cm^{-1} for the high-temperature phase and $+0.29 \text{ cm}^{-1}$ for the low-temperature phase. Indeed, a significant difference in the magnetic coupling can be found. However, they are still too weak to affect the high-temperature behavior of the measured magnetic properties.

- Fig 5: A reference system to understand how the M-L bonds are oriented should be provided as the energy variation of the d_{yz} orbital. The prime symbols should be explained.

Reply: Thank you for your suggestions. Our compound has a very low molecular symmetry (C_1). Thus, the five d orbitals are highly hybridized to form a reasonable set of ligand-field one-electron eigenfunctions. The composition coefficients are listed as follows:

The ligand field one electron eigenfunctions:

Orbital	Energy (ev)	Energy (cm ⁻¹)	d _{xy}	d _{yz}	d _{z2}	d _{xz}	d _{x2-y2}
d _{xy}	0	0	0.756175	0.257315	-0.49399	0.338958	-0.05539
d _{yz}	0.011	84.9	0.314253	-0.92254	0.113986	0.144533	-0.12763
d _{z2}	0.113	910.1	0.445166	0.181601	0.430838	-0.58357	-0.49262
d _{xz}	0.923	7447.1	0.235786	-0.14056	-0.21516	-0.6177	0.704839
d _{x2-y2}	1.09	8793.8	0.275093	0.173153	0.714888	0.376965	0.491095

In the manuscript, we use the convention to use the name of the d orbital that contributes most to the ligand-field eigenfunctions [*Nature Comm.* **9**, 2572 (2018)]. However, although it works well for some orbitals (e.g., d_{yz}), the reference frame cannot represent the appropriate orientation of other d lobes. Moreover, the reference frame will also exhibit rotation during the water-rotation process. Here, we show a tentative reference frame in Fig. 5e for better understanding.

On the basis of my concerns reported above I could recommend the publication in Nature Communications if all the above points will be accounted for and their replies fully justified.

Reviewer #2 (Remarks to the Author):

The manuscript by Sato et. al. describes a high-spin coordination complex of Co(II) with a coordinated water molecule, two coordinated nitrate ions, and a tridentate nitrogen donor ligand. In the solid state, this complex undergoes two structural phase transitions as function of temperature at ca. 113K and 157K. Through the intramolecular hydrogen bonding between the coordinated nitrate ions and the water molecule, these structural phase transitions bring about a step wise rotation of the coordinated water molecule, relative to the remaining coordination sphere, around the water-cobalt bond direction. The pi-bonding anisotropy of the coordinated water molecule leads to a concomitant change in the orientation of the magnetic susceptibility tensor.

The importance and novelty of this study is, according to the authors, deriving from it a) being the first experimental example of the importance of the anisotropic metal-water interaction for magnetic properties b) that the "reorientation of coordinated water can be controlled in crystals, ..." and "the orientation of coordinated water will be definitely considered when analyzing the performance of materials or devices in the presence of water molecules." This is a very nice study, it is well conducted, quite encompassing, and clearly reported. I think it is complete and suited for publication. I am, however, somewhat hesitant to recommend it for Nature Comms., mainly because i think that the impact and scope of appeal are not quite matching the high requirements of this journal. There are two primary arguments relating directly to how the importance of the study is presented in the manuscript. One concerns novelty and one concerns the impact in terms of providing an experimental handle on the properties of "materials or devices".

Addressing the novelty issue first:

I think that the claim that "Until now only theoretical studies have been conducted" (line 87), is incorrect. Several studies on water complexes of transition metal ions discuss the importance of the anisotropic donor properties of water and the inperplay with the electronic structure of the

metal center. This has led to classification of alums with classes exhibiting different water coordination modes (cf. e.g. Beattie, *et al.*, Dalton Trans, 1981, 2105 as well as Armstrong *et al.*, Dalton trans, 1983, 1973). Notably, the direct correlation between water coordination geometry and spectroscopic as well as magnetic properties have been analyzed in a series of papers by P.L.W. Tregenna-Piggott. Two examples: "Influence of the Mode of Water Coordination on the Electronic Structure of the $[V(OH_2)_6]^{3+}$ Cation", Journal of Solid State Chemistry 1992, 145, 460. and "Structure and Bonding of the Vanadium(III) Hexa-Aqua Cation. 1. Experimental Characterization and Ligand-Field Analysis" Inorg. Chem. 2004, 43, 8049. In the latter of these papers the influence of water coordination geometry on magnetic anisotropy is explicitly discussed. I think this deserves to be referenced, but also that it constitutes direct "prior art".

Reply: Thank you for your comments and valuable suggestions. We checked references 1–4 recommended by the reviewer in detail.

1. Structural Studies on the Caesium Alums, $CsM^{III}[SO_4]_2 \cdot 12H_2O$, Dalton Trans, 1981, 2105.

2. Crystal Structures of the Alums $CsM[SO_4]_2 \cdot 12H_2O$ ($M = Rh$ or Ir), Dalton trans., 1983, 1973.

3. Influence of the Mode of Water Coordination on the Electronic Structure of the $[V(OH_2)_6]^{3+}$ Cation, Journal of Solid State Chemistry 1992, 145, 460.

4. Structure and Bonding of the Vanadium(III) Hexa-Aqua Cation. 1. Experimental Characterization and Ligand-Field Analysis, Inorg. Chem. 2004, 43, 8049.

References 1 and 2 mainly introduce ten kinds of alum salts with different trivalent metal ions. These salts can be divided into two types (α and β) according to the coordination mode of water molecules. Reference 3 mainly discusses the energy level difference between the ground state and the excited state of compounds $RbV(SO_4)_2 \cdot 12H_2O$ and $Rb[Ga:V](SO_4)_2 \cdot 12H_2O$ (ca. 7% V) through the electronic absorption spectroscopy. These two compounds belong to α type and β type respectively. This shows that the coordination mode of water molecules affects the electronic structure. Reference 4 mainly discusses the difference in Hamiltonian parameters of the two types of hydrated vanadium ions in compound $[C(NH_2)_3][V(OH_2)_6](SO_4)_2$. The coordination modes of water in these two types of hydrated vanadium ions are different. The Hamiltonian parameters are obtained by characterizing the doped compound $[C(NH_2)_3][Ga:V(OH_2)_6](SO_4)_2$ via EPR.

According to references 3 and 4, the coordination mode of water will affect the electronic structure of hydrated vanadium ions. However, the coordination mode of water in the doped compounds, which were used for the EPR analysis, could not be accurately determined. We can only obtain statistical average positions of atoms through structure analysis in these doped compounds, because of the difference in radius between the host ion $[Ga^{3+}$ ($r = 0.62 \text{ \AA}$)] and the guest ion $[V^{3+}$ ($r = 0.64 \text{ \AA}$)]. Thus, it is impossible to accurately discuss the influence of the coordination mode of water molecules on the Hamiltonian parameters and magnetic anisotropy. Additionally, our work mainly discusses the effect of the reorientation of coordinated water with temperature on the magnetic anisotropy, rather than the effect of coordinated water in different systems with different coordination modes, as was discussed in references 3 and 4. In fact, coordinated water in different compounds with different coordination modes is a common phenomenon, whereas water molecule reorientation in response to external stimuli is challenging to achieve, especially in the densely packed crystalline phase. Hence, our case is different from the cases in references 3 and 4.

Considering the above, we made the following revisions to the article, including citing references 3 and 4.

However, the reorientation of water ligands through axial rotation around the metal–oxygen bonds, which might significantly affect the chemical and physical properties of metal complexes, has not been sufficiently evaluated^{11,12}.

11 Tregenna-Piggott, P. L. W., Spichiger, D., Carver, G. & Frey, B. Structure and Bonding of the Vanadium(III) Hexa-Aqua Cation. 1. Experimental Characterization and Ligand-Field Analysis. *Inorg. Chem.* **43**, 8049–8060 (2004).

12 Tregenna-Piggott, P. L. W., Best, S. P., Güdel, H. U., Weihe, H. & Wilson, C. C. Influence of the Mode of Water Coordination on the Electronic Structure of the $[V(OH_2)_6]^{3+}$ Cation. *J. Solid State Chem.* **145**, 460–470 (1999).

We hope you find our explanation satisfactory.

Concerning the consideration of the water coordination geometry in the context of materials and devices:

It is true that water - like many other ligands - have inherently anisotropic interactions with metal ions. Arguably, other pi-anisotropic ligands are more amenable than water to provide conformational control. Quantifying pi-anisotropies is indeed a very important step forward in understanding and analyzing electronic structure and magnetic properties. However, the quantum leap of controlling properties through conformational control lies well beyond the observation of phase transitions with associated conformational changes. Without the design strategy (cf. the statement "This study highlights that the rotational reorientation of coordinated water can be controlled in crystals") for generating the phase transitions, (cf. the statement "This study highlights that the rotational reorientation of coordinated water can be controlled in crystals", line 313) this remain, in my opinion, an interesting observation, but not control.

Reply: Thank you for your comments. As you commented, the generation of phase transitions is not always designable. However, according to our previous experience in the $[Co(1-bpp)(NO_3)_2]$ (1-bpp, 2,6-di(pyrazol-1-yl)pyridine) family of complexes (e.g. *Nature Commun.* **6**, 8810, 2015), potential structural phase transition is expected, mainly because that 1-bpp derivatives, as a rigid coplanar ligands, provide a relatively loose coordination environment for cobalt ions and the enough space for the rearrangement of other ligands, such as the nitrate ions and water ligands.

In addition, the water molecule is the smallest neutral ligand with a high polarity and hydrogen bonding capability, which causes them to readily reorient in response to changes in external environments, including the hydrogen-bonded networks. Considering these, we synthesized the compound presented here and focused on the orientation of the water ligand, which is mainly related to the adjacent hydrogen bonds.

Finally, we are sorry for our unclear statement. We have changed the statement "This study highlights that the rotational reorientation of coordinated water can be controlled in crystals" to "This study highlights that the reversible rotational reorientation of coordinated water in response to external stimuli, such as the temperature, can be realized in crystals."

Reviewer #3 (Remarks to the Author):

The manuscript "Water-oriented magnetic anisotropy transition" by Sheng-qun Su et al. describes a multi-technique study of a Co(II) complex. The authors are trying to ascribe the change of magnetic anisotropy of the Co(II) center ion to the rotation of the water ligand.

While in my opinion the findings are interesting and the experimental results seem to be of good quality I have major reservations from the magnetism point of view. The authors claim an easy-plane type magnetic anisotropy with a positive D value in the order of 30 cm⁻¹. This is not

consistent with the magnetic susceptibility data that they show in Figure 4b,d,f. Also, I am not convinced that the positive D value can be extracted with high certainty only from low-temperature HF-EPR measurements. With this D value, the higher lying $m_s = \pm 3/2$ states will not be populated, and a crossing with the $\pm 1/2$ states is not expected in experimental magnetic fields rendering the zfs inaccessible to HF-EPR at this low temperature.

Reply: Thank you for your comments. We are sorry that the x-, y-, and z-directions marked were not clear. We have relabeled three magnetic principal axes as X-, Y-, and Z-axes in Figure 4c. In Figure 4b,d,f, the x-, y-, and z-directions are defined in a single crystal used in magnetic measurements, where the x-direction is parallel to the a-axis, the y-axis is perpendicular to the (010) plane, and the z-axis is perpendicular to the xy plane.

It is true that the large D prohibits the visibility of the EPR transitions within the higher Kramer doublet or inter-Kramer transitions. However, due to the large spin-orbital coupling of Co(II) ions, forbidden transitions could be observed. We added Fig. S10 to show the relationship between EPR transitions and EPR spectra. Clearly, the case of $D > 0$ can fit the experimental spectrum better than that of $D < 0$.

Fig. 4 Determination of the magneto-crystalline anisotropy and its variations in single crystal 1. **a** The shape, faces, and the x-, y-, and z-directions of the single crystal used in magnetic measurements, where the x-direction is parallel to the a-axis, the y-axis is perpendicular to the (010) plane, and the z-axis is perpendicular to the xy plane. The relationship between the x-axis and the molecular orientation in the crystal is shown. **b** Temperature dependence of the $\chi_M T$ ($\chi_{Mx} T$, $\chi_{My} T$, and $\chi_{Mz} T$) values for single crystal 1. **c** Left: resonance field versus microwave frequency (quantum energy) for the EPR transitions of 1, where the green, blue, and red lines correspond to the simulations using the best fit spin Hamiltonian parameters with the magnetic field H parallel to the X-, Y-, and Z-axes of the ZFS tensor, respectively. The vertical dashed line represents the frequency (120 GHz). Right: The HF-EPR spectrum with its simulations at 4.2 K and 120 GHz. **d** Angular dependence of the magnetic susceptibility measured at 5 K and 190 K for the rotation along the x, y, and z axes, where the solid lines represent the calculated values. **f** Experimental and *ab initio* calculated

$\chi_{Mx}T$ curves in LTp and HTp. The changes in the calculated values after the phase transition are consistent with the experimental data.

Supplementary Fig. 10 HF-EPR spectra and energy levels as a function of magnetic field for the three canonical orientations of the field relative to the principal zfs axis at $T = 4.2$ K and $\nu = 120$ GHz. The spin Hamiltonian parameters used in the simulations were: $S = 3/2$, an axial g-tensor ($g_{(x)} = g_{(y)} = 2.45(2)$, $g_{(z)} = 2.25(2)$), $|D| = 32.01$ cm^{-1} , and $|E| = 4.30(2)$ cm^{-1} ($E/D \sim 0.13$).

Another point: The authors claim that they experimentally determine the water molecule rotation as a reason for the change of magnetic anisotropy, going beyond the case described in [G. Cucinotta et al., *Angew. Chem. Int. Ed.* 51, 1606 (2012)]. Concept-wise, calculations are still needed here to identify the water molecule rotation as a reason for the anisotropy change, among many other structural changes that occur during the phase transitions, which renders the present study less original than it is claimed by the authors.

Reply: Thank you for your comments. The work reported by Cucinotta et al. (*Angew. Chem. Int. Ed.* 51, 1606 (2012)) mainly predicted the effect of water orientation on the magnetic anisotropy using theoretical calculations. However, there is no experimental data except for the original molecular model. This is the main difference between that study and ours.

In our study, we observed an exciting two-step structural phase transition accompanied by a magnetic anisotropy change. Therefore, the structures of three phases with different water orientation was obtained, which is normally difficult in crystalline samples. Combined with the experimental results of the three phases and theoretical calculations, we confirmed that the reorientation of the water molecule is the main reason for the change in magnetic anisotropy.

Although, as the reviewer pointed out, the calculation is needed, we believe this study represents a major advance in understanding the water reorientation effects on magnetic anisotropy.

Other points:

1- The spin Hamiltonian model used in the work should be clearly noted down, e.g., in the SI.

Reply: Thank you for your suggestions. We have added the spin Hamiltonian model in the annotation in Supplementary Fig. 9 as follows. The solid lines represent the theoretical fittings using the PHI program by the anisotropic spin Hamiltonian (with $S = 3/2$ and $g_{(x)} = g_{(y)}$) as given in eq 1,³ $H = D(\hat{S}_z^2 - S(S+1)/3) + E(\hat{S}_x^2 - \hat{S}_y^2) + \mu_B g \hat{S} \cdot \hat{H}$ (1). The best-fit parameters are $g_{(x)} = g_{(y)} = 2.36$, $g_{(z)} = 2.71$, $D = 32.01 \text{ cm}^{-1}$, and $|E| = 0.28 \text{ cm}^{-1}$.

2- The authors should demonstrate that the spin-Hamiltonian model (D, E, g_x, g_y, g_z parameters) can reproduce the directional dependence along $\{x, y, z\}$ of the χT product shown in Fig 4b.

Reply: Thank you for your suggestions. Combined with the experimental Hamiltonian parameters $D, E, g_{(x)}, g_{(y)}$ and $g_{(z)}$ obtained from the HF-EPR analysis, the temperature-dependent anisotropic magnetic susceptibility that is in good agreement with the experimental value could be reproduced (Supplementary Fig. 7e). The deviation of the magnetic susceptibility along the y - and z -axis is mainly derived from the assumption that $g_{(x)} = g_{(y)}$.

Supplementary Fig. 7 (e) Experimental (black cycles) and simulated (red lines) temperature-dependent anisotropic magnetic susceptibility.

3- The assignment of the HF-EPR transitions is not consistent between Figs. 4c and S10

Reply: Thank you for your comments. We are sorry for this mistake. We have revised it.

4- Calculations: The authors should show the outcome of the ab-initio calculations for the χT of the single crystal (directional dependence along $\{x, y, z\}$, analogous to the experimental data shown in Fig 4).

Reply: Thank you for your suggestions. We have added the results of the *ab-initio* calculations for the magnetic susceptibility of the single crystal along the y - and z -directions to Supplementary Fig. 7d. The changing trend of the anisotropic magnetic susceptibility curves from the theoretical calculations is consistent with the experimental data. The deviation could be due to the vibronic coupling in the complex, temperature effect and error limits of experimental measurements and the water orientation from the XRD analysis.

Supplementary Fig. 7 (d). Experimental and *ab initio* calculated $\chi_{My}T$ and $\chi_{Mz}T$ curves in LTp and HTp.

5- The effective magnetic moment (μ_{eff}) is calculated to be 4.934 μ_B (page 5, Section "Anisotropic magnetic susceptibility transition. "). The authors should explain how such a large value is compatible with a S=3/2 system.

Reply: Thank you for your comments. Magnetic properties arise from the spin and the orbital angular momentum of electrons. In transition metal complexes, the orbital angular momentum is usually quenched by the crystal lattice symmetry. However, orbital angular momentum can be partially restored through spin-orbit coupling. We have added an explanation in the manuscript as "The large effective magnetic moment is induced by spin-orbit coupling which partially restores the quenched orbital angular momentum²⁷".

27 Bunting, P. C. *et al.* A linear cobalt(II) complex with maximal orbital angular momentum from a non-Aufbau ground state. *Science* **362**, 1378–1386 (2018).

6- I am not able to judge seriously the structure determination from XRD/NPD.

Reply: Thank you for your comments. We have further improved our structure models according to related comments and suggestions.

Reviewer #4 (Remarks to the Author):

This manuscript reports on the experimental and theoretical results of magnetic anisotropy influenced by water rotation in the Co-coordinating compound. The claimed experimental observation of water rotation and its effect on the anisotropy is significant. However, I don't think the claim on the structures is convincing enough at the current shape. In general, H atom could not be refined reliably by SHELXL (which is applied in this work) based on single-crystal X-ray diffraction data, especially for the reported data, all three datasets have a rather humble resolution (ca. 0.77 Å), considering the applied Mo radiation ($\lambda = 0.71075$ Å) at 70 and 140 K, and the Spring-8 high-resolution beamline with $\lambda = 0.41470$ Å at 190 K, much better resolution of the datasets should be obtainable. Probably, the authors cut off the high-resolution reflections, which is vital to the high accuracy measurement of the electron density, so critical to the determination of the H atoms in this case.

To alleviate the doubts on H determination, the authors tried neutron diffraction, however, with powder sample. I don't think powder neutron diffraction would give more reliable results in this case. Single-crystal neutron diffraction would be perfect here to determine the H and may

corroborate the claims made by the authors. The authors are strongly recommended to do this. However, considering the difficulties (probably not so in the research environment of the authors) to conduct single-crystal neutron diffraction, very high-resolution X-ray diffraction would be an alternative.

Reply: Thank you for your comments. In this case, the determination of the water orientation is important. We performed variable temperature single-crystal X-ray diffraction and powder neutron diffraction (deuterated sample). Because a large crystal is easy to break after the phase transition, we did not perform the single-crystal neutron diffraction experiments. In fact, powder neutron diffraction has been widely used to determine hydrogen positions, such as hydrogen positions in complex $\text{Co}(\text{acac})_2(\text{H}_2\text{O})_2$ (Nature Commun. 9, 2572, 2018). In particular, with the development of the latest generation of high-flux neutron powder diffractometers, operating under optimized collection geometries, they allow hydrogen positions to be extracted from the diffraction patterns of polycrystalline hydrogenous compounds without resorting to isotopic substitution (Chem Comm. 7, 2953–3124, 2009). According to the reviewer's suggestion, we performed high-resolution X-ray diffraction experiments this time using synchrotron radiation at beamline BL02B1 of SPring-8 to improve the accuracy of the positions of the hydrogen atoms. Unfortunately, we did not obtain perfect data for I_p and LT_p , which are important to accurately determine the positions of hydrogen atoms, because the breaking of the single crystal is inevitable after the first-order phase transition, especially when the crystal cannot be cooled slowly in a limited time. In addition, limited testing machine time is a limitation of our repeated measurement until obtaining a better data. However, we still observed a significant change in the orientation of water molecules with temperature. The HRXRD analysis show that the dihedral angle φ related to the angle of the water molecule changes from 89.17° at 190 K to 68.00° at 70 K, which is in good agreement with the XRD measurement results (from 85.30° at 190 K to 61.71° at 70 K), especially the results of the neutron diffraction measurements (from 88.50° at 194 K to 67.76° at 71 K). We hope that our explanation is satisfactory.

The main text was revised as follows.

Generally, single-crystal neutron diffraction is the preferred method to accurately determine the position of hydrogen atoms. However, single crystal 1 cannot be analyzed in this way because it easily breaks after the first-order phase transition as in many other switchable materials. To determine the unambiguous orientation of water, here, we use high-resolution single crystal diffraction (HRXRD) and powder neutron diffraction. The structure of 1 was analyzed with high-resolution single-crystal X-ray diffraction. The results show that the dihedral angle φ changes from 89.17° at 190 K to 68.00° at 70 K with almost the same variation (c.a. 21°) as in the XRD measurements. The bond lengths and angles between non-hydrogen atoms are almost unchanged compared with the XRD results (Supplementary Tables 1 and 2). In addition, the structures of the deuterated analog $[\text{Co}(\text{ONO}_2)_2(\text{D}_2\text{O})(\text{mprpz-d12})]$ (1-d14) were analyzed using variable-temperature single crystal XRD (120, 150, and 190 K) and powder neutron diffraction (71 K and 194 K) (Supplementary Fig. 6). The neutron diffraction data show that the dihedral angle φ related to the angle of the water molecule changes from 88.50° at 194 K to 67.76° at 71 K (Supplementary Table 1 and 4). This means that water rotates by approximately 21° between 71 K and 194 K, which is in good agreement with the XRD measurement results, especially the HRXRD results.

The high-resolution single-crystal X-ray diffraction analysis for complex 1 at 190 K (1_190K_hr), 140 K (1_140K_hr), and 70 K (1_70K_hr) were performed using synchrotron radiation at beamline BL02B1 of SPring-8 (Hyogo, Japan) with shorter wavelengths ($\lambda = 0.41195 \text{ \AA}$) using a Rigaku Mercury II detector. The crystal was enveloped in a temperature-controlled stream of dry

nitrogen gas (a mixture of nitrogen and helium gas) during data collection. The structures were solved by direct methods and were refined via full-matrix least-squares on F^2 using the SHELX software program with anisotropic thermal parameters for all non-hydrogen atoms. All the H-atoms were located using a difference Fourier map and refined freely. The configuration of the methyl groups in 1_190K_hr was refined with restraint DFIX. The X-ray crystallographic coordinates for the structures reported in this study have been deposited under deposition numbers CCDC 2058027-2058029.

Supplementary Table 1

	High-resolution single-crystal X-ray diffraction data for complex 1		
	70K	140 K	190 K
Formula	C ₁₂ H ₁₄ CoN ₈ O ₇	C ₁₂ H ₁₄ CoN ₈ O ₇	C ₁₂ H ₁₄ CoN ₈ O ₇
Formula weight	441.24	441.24	441.24
Crystal system	Triclinic	Triclinic	Triclinic
Space group	P -1	P -1	P -1
a (Å)	8.2771(2)	10.3326(1)	8.2326(1)
b (Å)	10.2100(2)	10.7744(2)	10.0620(2)
c (Å)	10.6512(2)	15.3301(2)	10.9956(2)
α (deg)	88.019(2)	95.4470(10)	87.0770(10)
β (deg)	83.032(2)	102.0980(10)	84.1150(10)
γ (deg)	67.162(2)	93.5840(10)	68.244(2)
V (Å ³)	823.36(3)	1655.26(4)	841.42(3)
Z	2	4	2
D _{calcd} (g cm ⁻³)	1.780	1.771	1.742
Wavelengths (Å)	0.4119	0.4119	0.4119
2 θ range for data collection (deg)	3.116-67.180	3.082-67.382	3.100-67.270
F (000)	450	900	450
R (int)	0.1414	0.1175	0.0662
GOF on F^2	1.086	1.041	1.095
R ₁ ^a [I > 2 σ (I)]	0.0895	0.1275	0.0714
ωR_2^b (all data)	0.2343	0.3921	0.2311

^a $R_1 = \sum ||F_o| - |F_c|| / \sum |F_o|$. ^b $\omega R_2 = \{ \sum [\omega(F_o^2 - F_c^2)^2] / \sum [\omega(F_o^2)^2] \}^{1/2}$

Except for the serious issue mentioned above, the other concerns are listed in the following: 1, the dihedral angle φ (defined between the plane defined by the N1, N3, and N5 atoms and the plane of the coordinated water molecule) is used to evaluate the water molecule rotation around the Co–O bond, please keep in mind, the Co–O bond is not on the defined N1, N3, and N5 plane.

Reply: Thank you for your comments. We chose the plane defined by three atoms (N1, N3, and N5) as the reference plane mainly for the following reasons, I) The coordination geometry of Co(II) ions is a distorted octahedron, and it will change after the phase transition. The plane defined by N1, N3 and N5 atoms can be regarded as the main plane of the molecule and is not easily affected by variations in the coordination environment because these three atoms belong to the same rigid conjugated ligand and participate in coordination. II) The Co-O(water) bond is close to this plane.

We have changed “The dihedral angle (φ) between the molecular plane (e.g. this plane defined by N1, N3, and N5 atoms) and the plane of the coordinated water molecule.....” to “The dihedral angle (φ) between the molecular plane (this plane defined by N1, N3, and N5 atoms is not easily affected by variations in the coordination environment and is close to the Co-O(water) bond) and the plane of the coordinated water molecule.....”

2, the supplementary crystal data of “1-d14” is not supplied. The data parameters presented in Supplementary Table 1 indicates that the structure at 120 K is already the low-temperature

phase of the proton cousin, as discussed in the manuscript, the deuterated compound shows a 3 K increase of phase transition, then it should be still at the intermediate phase at 120 K, similar to the 150 K, why not?

Reply: Thank you for your comments. As we mentioned in the manuscript, the phase transition temperature of deuterated sample 1-d2 is slightly higher (by approximately 3 K) than that of 1. However, the phase transition temperature of deuterated sample 1-d14 is increased by approximately 18 K from the magnetic data in Supplementary Fig. 8b. Thus, sample 1-d14 at 120 K is the low-temperature phase.

Supplementary Fig. 8b

3, the data at 70 K is a little strange to me since the low-temperature X-ray diffraction has lower thermal vibration, normally the data quality should be relatively higher than high-temperature data, at least not worse, it is worse than the 140 and 190 K datasets in this case. I recommend double-checking the data (especially for space group determination). For the missed data of “1-d14”, similar issues present for 120 and 150 K datasets. I can imagine one possibility: the phase transition is not completed. If this is the case, annealing during cooling down may help.

Reply: Thank you for your comments. From the variable-temperature X-ray diffraction data, the thermal vibration of atoms at 70 K is indeed lower. However, the data quality is usually dependent on the crystal quality. In this case, the crystal easily breaks after the first-order phase transition as in many other switchable materials. Perhaps this is the main reason for the difference in data quality between the HT phase and LT phase. It is also why we did not perform the single-crystal neutron diffraction.

4, the refinement of the H atom is not consistent (70 and 140 K: several H atoms were added as riding model, even for the critical water H atoms, the ADPs are refined as riding model; 190 K, several H atoms were added as riding model) for the three datasets, which is critical for the comparison. I recommend refining all the H atoms freely just as the authors did for the water H atoms at 190 K.

Reply: Thank you for your suggestions. We refined all the H-atoms again using the same method where all the H-atoms were located using a difference Fourier map and refined them freely. The new refinement results show that the water orientation (φ) and the potential

hydrogen bonds change slightly, which has been revised in the main text and in Supplementary Table 3.

5, the discussion on the phase transition from the structural perspective looks not so clear. I feel like the four nitrate ions nearby the water ligand are competing to generate H-bond with the water molecule.

Reply: Thank you for your comments. We have improved this section and added the following content to the main text.

Each water molecule may form hydrogen bonds with four adjacent nitrates. The orientation of water molecules is mainly related to the strongest hydrogen bonds O–H1WA(H2WA)···O and O–H1WB(H2WB)···O. When the phase transition produces changes in the nitrate's position, the water ligand correspondingly adjusts its orientation to maintain the shortest hydrogen bond interactions. The strongest hydrogen bonds change from O–H1WA···O3[-x+1, -y+2, -z+1] and O–H1WB···O6 at 190 K to O–H1WA···O5[-x+2, -y, -z] and O–H1WB···O3[-x+2, -y, -z+1] at 70 K.

6, the π - π interaction is not so clear, it looks like there is no real π - π interaction in the crystal lattice, since even the highlighted π - π interaction in Figure 1d doesn't show overlap on the π electron density, I would prefer to call it C–H- π interaction.

Reply: Thank you for your suggestion. We have checked the molecular packing and changed the " π - π interaction" to the "C–H- π interaction".

Fig. 1d The entire structure is mainly stabilized by O–H···O hydrogen bonds and C–H- π interactions.

7, it looks like indexing of the face roughly perpendicular to z-direction as 100 is a mistake in Figure 4a.

Reply: Thank you for your comments. We are sorry for this mistake. This face is the (001) face. We have revised it in Figure 4a.

a

Fig. 4a

REVIEWER COMMENTS

Reviewer #1 (Remarks to the Author):

Authors have significantly improved the paper taking the necessary steps to reply to my concerns.

In my opinion, the new results are sound but I do not fully agree with them when, related to the new data reported in Supplementary Table 6, they say that "These results suggest that the original model is reasonable for accounting for the observed phenomenon and can provide a correct semi-quantitative description of it. We hope that you find our explanation to be satisfactory."

The data obtained for the new model where the calculations were based on the H-optimized geometry, where two nitric acids forming H-bonds with a water molecule were considered (forth line in table) looks to me better than the original ones (second line in the table), i.e. the ones which they can directly compared with (i.e. active space constructed from the five 3d orbitals only).

For this reason, in my opinion, authors should take the new model as reference instead of the old one and, therefore, consider how the water rotation in this model affects the magnetic anisotropy. The effect of the larger active space (bonding and antibonding MO with d orbitals) should be include, too.

The inclusion of the above missing data will sure make the results and, therefore, the whole message of the paper much stronger.

Reviewer #3 (Remarks to the Author):

The authors have answered satisfactorily to most of my comments. Nevertheless, I would like the authors to address one question resulting from the simulations of the anisotropic magnetic susceptibility shown in the revised version in Supplementary Fig. 7e. The calculations are done for an easy-plane magnetic system (positive D). Yet, the anisotropic susceptibility looks like being of easy-axis type (negative D parameter). The following questions arise, which should be answered by the authors.

- 1) Is this apparent easy-axis nature related to the non-collinearity of the magnetic principal axes w.r.t the crystallographic directions?
- 2) As the spin Hamiltonian is formulated in the coordinate frame of the magnetic principal axes, was the transformation between these axes and the crystallographic axes properly taken into account in the susceptibility calculations?

I strongly recommend that the authors discuss these points in the main text to avoid misunderstandings. If the authors satisfactorily answer these questions and properly revise the manuscript I can recommend the work for publication in Nature Communications.

Reviewer #4 (Remarks to the Author):

The revised manuscript almost resolved my concerns on the previous version.

I appreciate the authors' efforts on the high-resolution single crystal X-ray diffraction data. There is still space to improve the data, even though not very critical. The authors are encouraged to double check the absorption correction on all three datasets. The rather large residual electron density nearby the heavy Co atom may be caused by improper absorption correction.

The revised description of the coordinated water molecule rotation in the phase transitions is still not satisfying for me. I think the current description of the water molecule rotation is oversimplified. The authors are encouraged to follow the description applied in the ref. 15 (Figure 1, Chem. Sci.10, 7233–7245 (2019)).

Additionally, it would be informative to prepare a supplementary table which clearly shows the structure parameters (specifically, the two angles characterizing the coordinated water molecule rotation, probably with addition of the Co-O(water) bond length/angle out of the N1-N3-N5 plane) obtained from different datasets. For this comparison, the atom numbering protocol should be consistent (NO₃ and H₂O ligands in the structures). By the way, Supplementary Table 2 could be re-edited to move the related structure parameters together for a straightforward comparison with addition of structure parameters determined from SC-XRD and NPD of 1-d14. The aim of these comparisons is clearly showing the structure information obtained with different diffraction methods, so giving an empirical evaluation of the uncertainty (every diffraction method would give an uncertainty evaluation its own, however, that evaluation is normally too small especially for H determination), the water rotation description could be described as rotation angle \pm uncertainty. By the way, I was supposed to recommend discussion of the structure based on the most reliable HR SC-XRD (probably removing the low-resolution datasets), since the currently obtained HR SC-XRD data quality is not overwhelmingly better (still much better in my opinion) than other methods, the current discussion is fine for me.

The size of atom displacement parameters (ADPs) of the two H atoms in water is different at low temperature/intermediate phases, especially significant for the 70 K (HR SC-XRD) data. This may be a clue of the stronger vibration or disorder of the H atom with large ADP. The authors are encouraged to make one-sentence comment on this to highlight the uncertainty of determination.

I am curious about the neutron powder diffraction results of the intermediate phase, if it happens that the authors already have the data, it would be very nice to bring it in. If not, it's not so critical, I do not recommend the authors to spend more time on it.

REVIEWER COMMENTS

Reviewer #1 (Remarks to the Author):

Authors have significantly improved the paper taking the necessary steps to reply to my concerns.

In my opinion, the new results are sound but I do not fully agree with them when, related to the new data reported in Supplementary Table 6, they say that "These results suggest that the original model is reasonable for accounting for the observed phenomenon and can provide a correct semi-quantitative description of it. We hope that you find our explanation to be satisfactory."

The data obtained for the new model where the calculations were based on the H-optimized geometry, where two nitric acids forming H-bonds with a water molecule were considered (forth line in table) looks to me better than the original ones (second line in the table), i.e. the ones which they can directly compared with (i.e. active space constructed from the five 3d orbitals only).

For this reason, in my opinion, authors should take the new model as reference instead of the old one and, therefore, consider how the water rotation in this model affects the magnetic anisotropy. The effect of the larger active space (bonding and antibonding MO with d orbitals) should be include, too.

The inclusion of the above missing data will sure make the results and, therefore, the whole message of the paper much stronger.

Response:

Thank you very much for your very careful inspection. We are so sorry for our mistake. As described in the Method section of our previous text, the model we used to study the water orientation effect is the partially optimized model where the positions of the hydrogen atoms were optimized with the inclusion of two guest nitrate anions. Indeed, after reading your comments, we are also surprising to notice the calculated magnetic parameters (line 2, effective g values of 3.603, 6.316, 2.080; and line 11, effective g values of 3.598, 6.319, 2.080) for these two models (^a Calculation based on the H-optimized geometry and active space constructed from the five 3d orbitals and ^f Calculation based on the H-optimized geometry including two nitric acid forming H-bonds with water molecule and active space constructed from the five 3d orbitals) are almost the same. Considering the significant dependence of magnetic properties on hydrogen positions, it would be very unreasonable. Therefore, we carefully check the structure and find the structures to obtain the two sets of data are almost identical except that the orientation of the water molecules differs by 0.4° (no difference has been found in the bond distances). They might be the same model that we optimized twice independently. Here we provide the overlapping molecular figures for these two models to check (Figure 1a). Considering these, we did not perform further calculations based on the molecular structure of model f because the results will be almost identical to the reported data. We just provided the correct data based on the model a (Figure 1b). It can be easily found that it is a completely different model; without the presence of two nitric acid molecules, the water molecule will rotate, forming the intermolecular hydrogen bonds with the coordinated nitrate groups during the structural optimization. The corresponding data have also been corrected by the appropriate model. We also check other models carefully to guarantee there are no additional mistakes in the supplementary table. To help readers to reproduce our results, atomic coordinates of

the correct optimized models are also provided as Supplementary Table 8. Thank you again for your insist on this issue, which helps us to avoid wrong results to be published.

Figure 1 (a) The molecular structure overlay; (b) the molecular structure of model a.

(previous version) **Supplementary Table 6** Principal components of the g tensor and the zero-field splitting parameters D and E .

	$(S = 3/2)$			$(S = 1/2)$			D (cm^{-1})	E (cm^{-1})
	g_x	g_y	g_z	g_x	g_y	g_z		
Exp. (HF-EPR)*	2.45	2.45	2.25	3.879	5.795	2.191	32.0	4.3
Calcd. (LTp) ^a	2.414	2.625	2.077	3.603	6.316	2.080	50.2	8.2
Calcd. (HTp) ^a	2.480	2.676	2.024	4.142	5.984	2.335	72.8	7.6
Calcd. (LTp) ^b	2.412	2.647	2.076	3.495	6.443	2.076	50.9	9.1
Calcd. (HTp) ^b	2.486	2.673	2.024	4.202	5.927	2.363	71.6	7.0
Calcd. (LTp) ^c	2.389	2.628	2.086	3.252	6.573	1.984	47.1	9.7
Calcd. (HTp) ^c	2.480	2.676	2.029	4.138	5.984	2.360	73.8	7.5
Calcd. (LTp) ^d	2.389	2.646	2.087	3.183	6.661	1.982	47.5	10.2
Calcd. (HTp) ^d	2.489	2.672	2.028	4.228	5.899	2.400	73.1	6.7
Calcd. (LTp) ^e	2.437	2.647	2.083	3.727	6.277	2.138	53.4	8.1
Calcd. (HTp) ^e	2.501	2.688	2.031	4.249	5.915	2.412	76.7	7.1
Calcd. (LTp) ^f	2.415	2.627	2.080	3.598	6.319	2.080	49.9	8.2
Calcd. (LTp) ^g	2.400	2.632	2.084	3.378	6.490	2.021	48.4	9.2

* Simultaneous fitting of the HF-EPR and magnetic susceptibility. g_x and g_y are restricted to be the same.

^a Calculation based on the H-optimized geometry and active space constructed from the five 3d orbitals.

^b Calculation based on the H-optimized geometry and active space constructed from the five 3d orbitals and five 4d orbitals as a double shell.

^c Calculation based on the H-fixed (according to the typical bond lengths obtained from neutron diffraction measurements and the X-ray determined bond angles) geometry and active space constructed from the five 3d orbitals.

^d Calculation based on the H-fixed (according to the typical bond lengths obtained from the neutron diffraction measurements and the X-ray determined bond angles) geometry and active space constructed from the five d orbitals and five 4d orbitals as a double shell.

^e Calculation based on the H-optimized geometry and active space constructed from the five 3d orbitals and their bonding partner orbitals.

^f Calculation based on the H-optimized geometry including two nitric acid forming H-bonds with water molecule and active space constructed from the five 3d orbitals.

^g Calculation based on the H-optimized geometry including two neighboring complex motifs forming H-bonds with water molecule and active space constructed from the five 3d orbitals.

(revised version) Supplementary Table 7 Principal components of the g tensor and the zero-field splitting parameters D and E .

	$(S = 3/2)$			$(S = 1/2)$			D (cm ⁻¹)	E (cm ⁻¹)
	g_x	g_y	g_z	g_x	g_y	g_z		
Exp. (HF-EPR) [*]	2.45	2.45	2.25	3.879	5.795	2.191	32.0	4.3
Calcd. (LTp) ^a	2.418	2.686	2.053	3.554	6.462	2.117	59.6	10.3
Calcd. (HTp) ^a	2.469	2.692	2.027	3.996	6.115	2.337	73.8	8.9
Calcd. (LTp) ^b	2.414	2.702	2.051	3.482	6.543	2.126	59.9	10.8
Calcd. (HTp) ^b	2.470	2.692	2.027	4.016	6.096	2.357	72.3	8.5
Calcd. (LTp) ^c	2.389	2.628	2.086	3.252	6.573	1.984	47.1	9.7
Calcd. (HTp) ^c	2.480	2.676	2.029	4.138	5.984	2.360	73.8	7.5
Calcd. (LTp) ^d	2.389	2.646	2.087	3.183	6.661	1.982	47.5	10.2
Calcd. (HTp) ^d	2.489	2.672	2.028	4.228	5.899	2.400	73.1	6.7
Calcd. (LTp) ^e	2.415	2.627	2.080	3.598	6.319	2.080	49.9	8.2
Calcd. (HTp) ^e	2.481	2.681	2.028	4.129	5.996	2.346	73.4	7.7
Calcd. (LTp) ^f	2.438	2.647	2.083	3.727	6.277	2.138	53.4	8.1
Calcd. (HTp) ^f	2.500	2.688	2.031	4.249	5.915	2.412	76.7	7.1
Calcd. (LTp) ^g	2.400	2.632	2.084	3.378	6.490	2.021	48.4	9.2

^{*} Simultaneous fitting of the HF-EPR and magnetic susceptibility. g_x and g_y are restricted to be the same.

^a Calculation based on the H-optimized geometry and active space constructed from the five 3d orbitals.

^b Calculation based on the H-optimized geometry and active space constructed from the five 3d orbitals and five 4d orbitals as a double shell.

^c Calculation based on the H-fixed (according to the typical bond lengths obtained from neutron diffraction measurements and the X-ray determined bond angles) geometry and active space constructed from the five 3d orbitals.

^d Calculation based on the H-fixed (according to the typical bond lengths obtained from the neutron diffraction measurements and the X-ray determined bond angles) geometry and active space constructed from the five d orbitals and five 4d orbitals as a double shell.

^e Calculation based on the H-optimized geometry including two nitric acid forming H-bonds with water molecule and active space constructed from the five 3d orbitals.

^f Calculation based on the H-optimized geometry and active space constructed from the five 3d orbitals and their bonding partner orbitals.

^g Calculation based on the H-optimized geometry including two neighboring complex motifs forming H-bonds with water molecule and active space constructed from the five 3d orbitals.

Reviewer #3 (Remarks to the Author):

The authors have answered satisfactorily to most of my comments. Nevertheless, I would like the authors to address one question resulting from the simulations of the anisotropic magnetic susceptibility shown in the revised version in Supplementary Fig. 7e. The calculations are done for an easy-plane magnetic system (positive D). Yet, the anisotropic susceptibility looks like being of easy-axis type (negative D parameter). The following questions arise, which should be answered by the authors.

1) Is this apparent easy-axis nature related to the non-collinearity of the magnetic principal axes w.r.t the crystallographic directions?

Response:

Yes. Indeed, the angular dependent measurements were performed on directions defined based on the crystalline faces, which form large angles with respect to the magnetic principal axes (as shown in Figure 2, the hard-axis of the magnetic principal axes system (Z-axis) is close to the xy plane of the single crystal), leading to the observed easy-axis nature.

Figure 2. The relationship between the directions of the applied magnetic field (x-, y- and z-axis of the single crystal) and the magnetic principal axes (X-, Y- and Z-axis) at the low- and high-temperature phase.

2) As the spin Hamiltonian is formulated in the coordinate frame of the magnetic principal axes, was the transformation between these axes and the crystallographic axes properly taken into account in the susceptibility calculations?

Response:

Related to the response to question 1, when we were calculating the anisotropic magnetic susceptibility of the crystal, we properly considered the relationship between the magnetic principal axes and the crystallographic axes and make the transformation. In the theoretical

calculations, we calculated the temperature-dependent susceptibility tensors in the molecular frames (defining x' -axis along the crystalline a -axis and y' -axis along the direction that lies in the ab -plane and is perpendicular to the a -axis, and z' -axis along the direction perpendicular to both x' - and y' -axis). This frame differs from the measurement xyz -axis system by only 0.85 degree for LT phase and 0.72 degree for HT phase, and the corresponding coordinates transformation matrices were used to rotate the susceptibility tensors to the correct direction. When dealing with the spin Hamiltonian, we introduced the azimuth angles (θ and ϕ , as shown in Figure 2 above) to determine the magnetic field direction to simulate the susceptibilities.

According to your suggestion, we added the following sentences in the text:

Here, the anisotropic susceptibility with apparent easy-axis nature is related to the non-collinearity of the magnetic principal axes (X -, Y - and Z -axis) with the crystallographic directions (x -, y - and z -axis). When we were calculating the anisotropic magnetic susceptibility of the crystal, we properly considered the relationship between the magnetic principal axes and the crystallographic axes and make the transformation.

I strongly recommend that the authors discuss these points in the main text to avoid misunderstandings. If the authors satisfactorily answer these questions and properly revise the manuscript I can recommend the work for publication in Nature Communications.

Response:

Thank you very much for your comments and suggestions. According to your suggestion, we added the following sentence in the text.

Here, the anisotropic susceptibility with apparent easy-axis nature is related to the non-collinearity of the magnetic principal axes (X -, Y - and Z -axis) with the crystallographic directions (x -, y - and z -axis). When we were calculating the anisotropic magnetic susceptibility of the crystal, we properly considered the relationship between the magnetic principal axes and the crystallographic axes and make the transformation.

Reviewer #4 (Remarks to the Author):

The revised manuscript almost resolved my concerns on the previous version.

I appreciate the authors' efforts on the high-resolution single crystal X-ray diffraction data. There is still space to improve the data, even though not very critical. The authors are encouraged to double check the absorption correction on all three datasets. The rather large residual electron density nearby the heavy Co atom may be caused by improper absorption correction.

Response:

Thank you very much for your comments. In this work, the empirical absorption correction, which was performed according to the built-in intelligent decision making based on current dataset statistics of CrysAlis Pro software, was used on all three datasets. We have double checked the absorption correction on them and tried to make empirical absorption correction

again on them by changing the absorption coefficient (although it is usually obtained by calculating from molecular formula), but we did not obtain the better results. Usually, the numerical correction is the other commonly used correction method. However, we could not use this method in our cases because the crystal shape could not be determined when the single crystal sample was measured in Spring-8.

The revised description of the coordinated water molecule rotation in the phase transitions is still not satisfying for me. I think the current description of the water molecule rotation is oversimplified. The authors are encouraged to follow the description applied in the ref. 15 (Figure 1, Chem. Sci.10, 7233–7245 (2019)).

Additionally, it would be informative to prepare a supplementary table which clearly shows the structure parameters (specifically, the two angles characterizing the coordinated water molecule rotation, probably with addition of the Co-O(water) bond length/angle out of the N1-N3-N5 plane) obtained from different datasets. For this comparison, the atom numbering protocol should be consistent (NO₃ and H₂O ligands in the structures). By the way, Supplementary Table 2 could be re-edited to move the related structure parameters together for a straightforward comparison with addition of structure parameters determined from SC-XRD and NPD of 1-d14. The aim of these comparisons is clearly showing the structure information obtained with different diffraction methods, so giving an empirical evaluation of the uncertainty (every diffraction method would give an uncertainty evaluation its own, however, that evaluation is normally too small especially for H determination), the water rotation description could be described as rotation angle \pm uncertainty. By the way, I was supposed to recommend discussion of the structure based on the most reliable HR SC-XRD (probably removing the low-resolution datasets), since the currently obtained HR SC-XRD data quality is not overwhelmingly better (still much better in my opinion) than other methods, the current discussion is fine for me.

Response:

Thank you very much for your valuable suggestions. According to your suggestion, to better describe the orientation of the water, the other parameter φ_1 , which is the angle between the Co-O(water) bond and the molecular plane of water, is introduced. The value of the angle between the Co-O(water) bond and the molecular plane (φ_0) was also added in the text. These structure parameters including φ , φ_0 , φ_1 , ψ and ω from different datasets are list in the Supplementary Table 4 to clearly show the changes of these main structural parameters. We have revised the CIF files of data 1_70K_hr and 1_D14_120K with the consistent atom numbering protocol for this comparison. In addition, Supplementary Table 2 was re-edited by moving the related structure parameters together and adding the data of 1-d14. From four sets of data, especially data of 1 (HRXRD) and 1-d₁₄ (PND)), the water reorientation is mainly attained by the rotation of water molecules around the Co–O bond from HTp to LTp, in which the angle φ changes by $21.32 \pm 3.67^\circ$.

Supplementary Table 4 The structure parameters related to the orientation of the water and nitrates from the different phases of complex 1 and 1-d₁₄.

		φ (°)	φ_1 (°)	φ_0 (°)	ψ (°)	ω (°)
Hp	SCXRD-1	85.30	27.56	0.65	133.53	159.64
	SCXRD-1(hr)	93.31	15.11	0.57	133.41	159.55

	SCXRD-1_d14	83.31	22.55	0.57	133.59	159.63
	NPD-1_d14	88.50	38.67	4.04	137.61	162.31
Ip	SCXRD-1	70.66 / 68.19	30.88 / 21.49	2.22 / 6.07	133.49 / 123.80	153.03 / 141.33
	SCXRD-1(hr)	75.01 / 58.35	58.75 / 25.92	2.13 / 6.07	133.53 / 123.93	153.32 / 141.23
	SCXRD-1_d14	71.79 / 69.48	29.97 / 20.77	2.30 / 6.12	134.04 / 123.28	151.45 / 140.85
	NPD-1_d14					
Lp	SCXRD-1	61.71	21.22	9.55	126.38	141.35
	SCXRD-1(hr)	68.00	15.17	9.31	126.47	141.24
	SCXRD-1_d14	67.70	22.78	9.03	126.22	141.43
	NPD-1_d14	67.73	36.26	12.19	129.51	141.75

φ , angle between the molecular plane and the plane of the coordinated water molecule;

φ_0 , angle between the Co-O(water) bond and the molecular plane (defined by atoms N1, N3 and N5);

φ_1 , angle between the Co-O(water) bond and the plane of the coordinated water molecule;

ψ , the dihedral angle N3-Co1-O1-O3;

ω , the dihedral angle N3-Co1-O4-O6.

Supplementary Table 2 Selected bond distances (Å) and angles (°) from the different phases of complex **1** and **1-d₁₄**.

	Hp			
	SCXRD-1	SCXRD-1(hr)	SCXRD-1_d14	NPD-1_d14
Co1-O1W	2.0136(12)	2.0178(6)	2.0048(19)	2.0149(14)
Co1-O1	2.0981(11)	2.0968(6)	2.0900(16)	2.098(24)
Co1-O4	2.1352(11)	2.1348(6)	2.1304(17)	2.1350(16)
Co1-N1	2.1865(13)	2.1749(5)	2.181(2)	2.1871(13)
Co1-N3	2.1105(11)	2.1147(4)	2.1032(17)	2.1131(13)
Co1-N5	2.1748(12)	2.1924(6)	2.1706(18)	2.1770(15)
O1w-Co1-O1	93.28(5)	93.37(3)	93.38(8)	93.04(6)
O1w-Co1-N3	175.58(5)	175.44(3)	175.43(8)	173.58(7)
O1-Co1-N3	88.82(4)	88.64(2)	88.75(7)	89.32(6)
O1w-Co1-O4	90.16(5)	90.18(3)	90.05(8)	93.36(6)
O1-Co1-O4	169.91(5)	169.95(3)	170.04(7)	173.05(8)
N3-Co1-O4	87.13(4)	87.19(2)	87.31(6)	89.70(5)
O1w-Co1-N5	102.91(6)	110.28(4)	102.72(9)	99.90(6)
O1-Co1-N5	84.61(4)	97.09(3)	84.62(7)	85.19(7)
N3-Co1-N5	73.40(4)	73.48(2)	73.43(7)	74.56(5)
O4-Co1-N5	85.37(5)	90.48(3)	85.49(7)	87.92(6)
O1w-Co1-N1	109.84(6)	102.83(4)	109.87(9)	108.08(6)
O1-Co1-N1	97.31(5)	84.69(2)	97.42(7)	95.78(7)
N3-Co1-N1	73.70(5)	73.254(17)	73.81(7)	77.49(5)
O4-Co1-N1	90.42(5)	85.36(3)	90.25(8)	90.72(6)
N5-Co1-N1	146.99(5)	146.63(2)	147.12(7)	152.02(7)
	Ip			
	SCXRD-1	SCXRD-1	SCXRD-1_d14	
Co1-O1W	2.0146(14)	2.0199(14)	2.0199(14)	
Co1-O1	2.1057(13)	2.1048(12)	2.1048(12)	
Co1-O4	2.1165(12)	2.1155(12)	2.1155(12)	
Co1-N1	2.2032(17)	2.2001(15)	2.2001(15)	
Co1-N3	2.1181(15)	2.1166(11)	2.1166(11)	
Co1-N5	2.1766(16)	2.1722(12)	2.1722(12)	

O1w-Co1-O1	91.93(6)	92.03(6)	92.03(6)	
O1w-Co1-N3	172.46(6)	172.52(7)	172.52(7)	
O1-Co1-N3	87.63(6)	87.58(5)	87.58(5)	
O1w-Co1-O4	93.22(6)	92.83(5)	92.83(5)	
O1-Co1-O4	171.85(5)	171.87(6)	171.87(6)	
N3-Co1-O4	86.44(6)	86.74(5)	86.74(5)	
O1w-Co1-N5	99.33(6)	99.18(6)	99.18(6)	
O1-Co1-N5	87.42(6)	87.34(5)	87.34(5)	
N3-Co1-N5	73.13(6)	73.23(4)	73.23(4)	
O4-Co1-N5	85.51(6)	85.45(6)	85.45(6)	
O1w-Co1-N1	113.82(6)	114.00(7)	114.00(7)	
O1-Co1-N1	94.35(6)	94.32(6)	94.32(6)	
N3-Co1-N1	73.72(6)	73.58(5)	73.58(5)	
O4-Co1-N1	89.35(6)	89.65(6)	89.65(6)	
N5-Co1-N1	146.69(6)	146.67(5)	146.67(5)	
Co2-O2	2.0344(14)	2.0338(13)	2.0338(13)	
Co2-O7	2.1073(13)	2.1048(12)	2.1048(12)	
Co2-O10	2.1092(12)	2.1051(13)	2.1051(13)	
Co2-N9	2.1597(17)	2.1618(14)	2.1618(14)	
Co2-N11	2.1133(15)	2.1041(14)	2.1041(14)	
Co2-N13	2.1646(17)	2.1621(15)	2.1621(15)	
O2w-Co2-O7	90.41(6)	90.61(6)	90.61(6)	
O2w-Co2-N11	174.97(6)	175.16(6)	175.16(6)	
O7-Co2-N11	86.62(6)	86.42(5)	86.42(5)	
O2w-Co2-O10	94.32(6)	94.09(6)	94.09(6)	
O7-Co2-O10	168.47(5)	168.60(6)	168.60(6)	
N11-Co2-O10	87.87(6)	88.15(6)	88.15(6)	
O2w-Co2-N13	102.05(6)	101.91(6)	101.91(6)	
O7-Co2-N13	85.07(6)	85.10(5)	85.10(5)	
N11-Co2-N13	73.65(7)	74.04(7)	74.04(7)	
O10-Co2-N13	83.75(6)	83.77(6)	83.77(6)	
O2w-Co2-N9	110.61(6)	110.09(6)	110.09(6)	
O7-Co2-N9	98.17(6)	97.94(6)	97.94(6)	
N11-Co2-N9	73.88(6)	74.16(6)	74.16(6)	
O10-Co2-N9	90.01(6)	90.19(7)	90.19(7)	
N13-Co2-N9	147.13(6)	147.79(6)	147.79(6)	
Lp				
	SCXRD-1	SCXRD-1	SCXRD-1_d14	NPD-1_d14
Co1-O1W	2.025(2)	2.0213(5)	2.0140(18)	2.0220(41)
Co1-O1	2.1214(19)	2.1216(6)	2.108(2)	2.1210(45)
Co1-O4	2.0994(19)	2.1034(6)	2.093(2)	2.0989(49)
Co1-N1	2.159(2)	2.1601(7)	2.159(2)	2.158(11)
Co1-N3	2.117(2)	2.1164(6)	2.101(2)	2.1036(22)
Co1-N5	2.159(2)	2.1609(6)	2.154(2)	2.1457(94)
O1w-Co1-O1	90.89(8)	90.98(2)	90.79(8)	88.21(5)
O1w-Co1-N3	171.77(8)	171.93(3)	172.12(8)	168.75(6)
O1-Co1-N3	84.08(8)	84.08(2)	84.46(8)	83.36(5)
O1w-Co1-O4	95.33(8)	95.16(3)	95.20(8)	98.53(8)
O1-Co1-O4	170.93(8)	170.88(2)	171.00(8)	173.26(7)
N3-Co1-O4	88.97(8)	89.03(2)	88.81(8)	89.94(5)
O1w-Co1-N5	99.80(8)	99.96(3)	100.11(8)	97.82(5)
O1-Co1-N5	88.85(8)	88.65(2)	88.51(8)	91.23(5)
N3-Co1-N5	73.66(9)	73.80(3)	73.55(9)	75.00(6)

O4–Co1–N5	83.59(8)	83.65(2)	83.84(8)	87.83(9)
O1w–Co1–N1	113.15(8)	113.10(3)	112.84(8)	112.04(8)
O1–Co1–N1	94.72(8)	94.72(3)	94.96(8)	90.93(12)
N3–Co1–N1	73.87(9)	73.58(3)	73.93(9)	75.66(5)
O4–Co1–N1	88.92(9)	89.08(3)	88.91(9)	86.59(6)
N5–Co1–N1	146.75(9)	146.66(2)	146.78(9)	150.12(13)

The size of atom displacement parameters (ADPs) of the two H atoms in water is different at low temperature/intermediate phases, especially significant for the 70 K (HR SC-XRD) data. This may be a clue of the stronger vibration or disorder of the H atom with large ADP. The authors are encouraged to make one-sentence comment on this to highlight the uncertainty of determination.

Response:

Thank you very much! According to your comments, we have added the following content in the text, "However, there is still uncertainty about the orientation of water molecules because of the vibration or disorder of H atoms, which could be obtained from the different atom displacement parameters of the H atoms in water at Lp and Ip, especially for the 70 K (HRXRD) data".

I am curious about the neutron powder diffraction results of the intermediate phase, if it happens that the authors already have the data, it would be very nice to bring it in. If not, it's not so critical, I do not recommend the authors to spend more time on it.

Response:

Thank you very much! We have not performed the neutron powder diffraction for the Ip of our sample because we have not obtained enough measurement time.

REVIEWER COMMENTS

Reviewer #1 (Remarks to the Author):

The paper has been correctly amended and, therefore, I recommend the paper for the publication in the present form.

Reviewer #3 (Remarks to the Author):

The authors have fully addressed my concerns and revised the manuscript accordingly. Hence from my side there are no further issues, and I recommend the manuscript for publication.

Reviewer #4 (Remarks to the Author):

The revised manuscript resolved my concerns about the previous version. There are minor revision suggestions on the current version:

The coordinated water rotation angle between HT-LT phases (Hp-Lp in the manuscript) is claimed as $21.32 \pm 3.67^\circ$, I think typing as $21 \pm 4^\circ$ is more sensible. The details of the suggestion/evaluation of this number should be interpreted in the supplementary information. It seems that the authors made simple statistics with the four measurements to get the average $\Delta\varphi$ and uncertainties without considering the weighting factor of the measurement method, which is not adequate in my opinion. The datasets acquired with four measurements are not equally weighted, the NPD and SCXRD of 1-d14 are from deuterated crystal 1-d14, as shown in the manuscript, deuteration is not fully innocent on the water rotation behaviour, so the comparison between NPD of crystal 1-d14 and SCXRD of crystal 1 should be cautious. I expected the authors to make a sensible evaluation of the data acquired from the four measurements, with HR-SCXRD of crystal 1 as the most reliable data. I should say more here, the uncertainty of measurements is not a terrible thing to avoid mentioning in the manuscript, it helps to properly evaluate the experimental results. There is no question mark on the rotation of the coordinated water molecule in the temperature-driven phase transition reported in this manuscript, I believe that a rigorous evaluation/description of the coordinated water molecule rotation is very important. There are discrepancies between manuscript and SI on the φ angle of 190 K-hr data. In the manuscript, it is 89.17° , which is 93.31° in the SI. I checked the submitted data, it is $89.17/90.83^\circ$. The authors are recommended to check all the angles very carefully, especially for the consistency of defining the φ angle, which is critical to determine the dihedral angles, such as the above mentioned $89.17/90.83^\circ$.

The authors are recommended to carefully update the crystal structure/data table in the supplementary information. Such as, for the crystal data of 1-190 K (Supplementary Table 1), R1 is 0.0305, not 0.0309 reported in the current table.

The description of the coordinated water rotation in the manuscript is a little messy. I recommend the authors move the details to SI to increase the readability of the manuscript for wider audiences.

REVIEWER COMMENTS

Reviewer #1 (Remarks to the Author):

The paper has been correctly amended and, therefore, I recommend the paper for the publication in the present form.

Response:

Thank you very much.

Reviewer #3 (Remarks to the Author):

The authors have fully addressed my concerns and revised the manuscript accordingly. Hence from my side there are no further issues, and I recommend the manuscript for publication.

Response:

Thank you very much.

Reviewer #4 (Remarks to the Author):

The revised manuscript resolved my concerns about the previous version. There are minor revision suggestions on the current version:

The coordinated water rotation angle between HT-LT phases (Hp-Lp in the manuscript) is claimed as $21.32 \pm 3.67^\circ$, I think typing as $21 \pm 4^\circ$ is more sensible. The details of the suggestion/evaluation of this number should be interpreted in the supplementary information. It seems that the authors made simple statistics with the four measurements to get the average $\Delta\phi$ and uncertainties without considering the weighting factor of the measurement method, which is not adequate in my opinion. The datasets acquired with four measurements are not equally weighted, the NPD and SCXRD of 1-d14 are from deuterated crystal 1-d14, as shown in the manuscript, deuteration is not fully innocent on the water rotation behaviour, so the comparison between NPD of crystal 1-d14 and SCXRD of crystal 1 should be cautious. I expected the authors to make a sensible evaluation of the data acquired from the four measurements, with HR-SCXRD of crystal 1 as the most reliable data. I should say more here, the uncertainty of measurements is not a terrible thing to avoid mentioning in the manuscript, it helps to properly evaluate the experimental results. There is no question mark on the rotation of the coordinated water molecule in the temperature-driven phase transition reported in this manuscript, I believe that a rigorous evaluation/description of the coordinated water molecule rotation is very important.

There are discrepancies between manuscript and SI on the ϕ angle of 190 K-hr data. In the manuscript, it is 89.17° , which is 93.31° in the SI. I checked the submitted data, it is $89.17/90.83^\circ$. The authors are recommended to check all the angles very carefully, especially for the consistency of defining the ϕ angle, which is critical to determine the dihedral angles, such as the above mentioned $89.17/90.83^\circ$.

The authors are recommended to carefully update the crystal structure/data table in the supplementary information. Such as, for the crystal data of 1-190 K (Supplementary Table 1), R1 is 0.0305, not 0.0309 reported in the current table.

The description of the coordinated water rotation in the manuscript is a little messy. I

recommend the authors move the details to SI to increase the readability of the manuscript for wider audiences.

Response:

Thank you very much for your comments and suggestions.

As commented, the deuteration is not fully innocent on the water rotation behavior and HR-SCXRD of crystal 1 is the most reliable data. Therefore, in the revised version, we directly adopted the variation of the angle φ of $21.2\pm 0.2^\circ$ from HTp to LTp based on HRXRD data. In addition, we have included the average value and estimated deviation of the variation of the angle φ of $22\pm 1^\circ$ from data SCXRD-1 and SCXRD-1(hr) in the SI. The difference between them lies within the experimental error bar.

Besides, we have corrected and updated all the structural parameters and crystallographic data listed in the manuscript and supplementary information to make them consistent with the data in the CIF files.

Finally, to improve the readability and clarity of our manuscript, we have moved the following content, including the description and analysis of HRXRD data of 1 and SCXRD data and PND data of 1-d14 to the SI.

Supplementary Methods

Determination of water orientation. To determine the unambiguous orientation of water, here, we use high-resolution single crystal diffraction (HRXRD) and powder neutron diffraction (PND). Generally, single-crystal neutron diffraction is the preferred method to accurately determine the position of hydrogen atoms. However, single crystal 1 cannot be analyzed in this way because it easily breaks after the first-order phase transition. The structure of 1 was analyzed with high-resolution single-crystal X-ray diffraction. The results show that the dihedral angle φ changes from 89.17° at 190 K to 68.00° at 70 K with almost the same variation (21.2°) as that obtained in the XRD measurements, while the angle φ_1 only changes by 0.06° . The bond lengths and angles between non-hydrogen atoms are almost unchanged compared with the XRD results (Supplementary Tables 1 and 2). However, there is still uncertainty about the orientation of water molecules because of the vibration or disorder of H atoms, which could be obtained from the different atom displacement parameters of the H atoms in water at Lp and Ip, especially for the 70 K (HRXRD) data. In addition, the structures of the deuterated analog [Co(ONO₂)₂(D₂O)(mprpz-d₁₂)] (1-d14) were analyzed using variable-temperature single crystal XRD (120, 150, and 190 K) and powder neutron diffraction (71 K and 194 K) (Supplementary Fig. 6). From the single crystal XRD data, the angle φ changes by 15.61° from 83.31° at 190 K to 67.70° at 120 K, while the angle φ_1 only changes by 0.23° . The neutron diffraction data show that the dihedral angle φ is related to the angle of the water molecule and changes from 88.50° at 194 K to 67.73° at 71 K, while the angle φ_1 only changes by 2.41° (Supplementary Table 1 and 5).

From four sets of data, the water reorientation in complex 1 is mainly attained by the rotation of water molecules around the Co–O bond after the structural transition. The variation of the angle φ in complex 1 from HTp to LTp is described in the text with the result of HRXRD that is $21.2\pm 0.2^\circ$ (Supplementary Methods and Table 4). Herein, we include the average value and estimated deviation of the variation of the angle φ of $22\pm 1^\circ$ from data SCXRD-1 and SCXRD-1(hr). The difference between them lies within the experimental error bar.

REVIEWER COMMENTS

Reviewer #4 (Remarks to the Author):

The authors clarified all my concerns. I am happy to recommend it for publication in the current shape.

REVIEWERS' COMMENTS

Reviewer #4 (Remarks to the Author):

The authors clarified all my concerns. I am happy to recommend it for publication in the current shape.

Response:

Thank you very much for your comments and suggestions.